DESY-25-098

# $\mathcal{CP}$-Analyses with Symbolic Regression

Henning Bahl[1], Elina Fuchs[2,3,4], Marco Menen[2,3], and Tilman Plehn[1,5]

**1** Institut für Theoretische Physik, Universität Heidelberg, Philosophenweg 16, 69120 Heidelberg, Germany
**2** Institut für Theoretische Physik, Leibniz Universität Hannover, Appelstraße 2, 30167 Hannover, Germany
**3** Physikalisch-Technische Bundesanstalt, Bundesallee 100, 38116 Braunschweig, Germany
**4** Deutsches Elektronen-Synchrotron DESY, Notkestr. 85, 22607 Hamburg, Germany
**5** Interdisciplinary Center for Scientific Computing (IWR), Universität Heidelberg, Germany

## Abstract

**Searching for $\mathcal{CP}$ violation in Higgs interactions at the LHC is as challenging as it is important. Although modern machine learning outperforms traditional methods, its results are difficult to control and interpret, which is especially important if an unambiguous probe of a fundamental symmetry is required. We propose solving this problem by learning analytic formulas with symbolic regression. Using the complementary PySR and SymbolNet approaches, we learn $\mathcal{CP}$-sensitive observables at the detector level for WBF Higgs production and top-associated Higgs production. We find that they offer advantages in interpretability and performance.**

# 1 Introduction

More than ten years after the discovery of a SM-like Higgs boson [1,2], many questions about its nature remain unanswered [3]. A particularly interesting property of the Higgs boson is its $\mathcal{CP}$ nature. Any deviation from a fully $\mathcal{CP}$-even coupling of the Higgs to other SM particles would constitute a clear signal of BSM physics and could give us insight into how the matter-antimatter asymmetry observed in the Universe is generated [4,5]. Although the rapidly growing LHC datasets allow for increasingly precise measurements, they also pose challenges for experimental analyses. First, data analysis techniques need to process huge datasets in a reasonable amount of time. At the same time, they have to work with sparse amounts of data to identify rare processes against the large background and study their properties. Finally, analyses need to cover large portions of phase space and BSM parameter space.

In recent years, various modern machine learning (ML) approaches have been developed to address these problems, offering an excellent compromise between speed, accuracy, and data efficiency. One goal is to construct an observable that gives optimal sensitivity [6–8] to, for instance, $\mathcal{CP}$ violation in the Higgs sector. At parton level, the Neyman-Pearson lemma [9] gives an exact definition of the optimal observable. However, it is much harder to define once we include parton showering, detector resolution, and uncertainties in the reconstruction algorithms [10]. This issue can be approached using ML methods with very convincing results [11–19]. These powerful numerical methods do not lead to an analytic expression, so formulas for reco-level optimal observable are only known in rare cases, such as $p_{T,j_1} p_{T,j_2} \sin(\Delta\phi_{jj})$ in the context of VBF production [20].

In addition to pure performance, the interpretability and control of ML-approaches is important in particular when it comes to probing fundamental symmetries. Interpretability can be achieved in a number of ways. For example, Shapley values allow to extract the relative importance of input parameters on ML-outputs [17, 21, 22]. However, they fall short of analytic equations. Symbolic regression (SR) allows us to extract the shape of a function and its parameters from data. In addition to theoretical insights, a single equation is fast to evaluate, and it simplifies reinterpretation. It is especially interesting in the context of symmetries as their conservation — or violation — will be imprinted in the function. In spite of its potential, SR has only been sparsely used in LHC physics so far [20, 23–30].

In this work, we show how SR can be used for detecting $\mathcal{CP}$ violation in the Higgs sector. In particular, we demonstrate how it can be used to construct optimal $\mathcal{CP}$-odd observables and to reconstruct parton-level $\mathcal{CP}$-sensitive observables from reconstruction-level data. Throughout the paper, we compare results derived using two complementary SR-approaches. First, we use PySR [31], which is based on the concept of evolutionary algorithms. Second, we use an enhanced version of SymbolNet [32], which relies on a sparsely connected neural network. After training, the NN resembles a fixed analytic equation that can be extracted. Alternative tools include AI Feynman with its improved reconstruction thanks to the recursive simplification and LASR [33], which uses large language models to accelerate SR.

In Section 2, we first introduce the concept of SR and the functionalities of PySR and SymbolNet in detail. Then, we employ both algorithms to find an optimal $\mathcal{CP}$-odd observable in VBF Higgs production in Section 3. Afterwards, Section 4 deals with reconstructing the $\mathcal{CP}$-sensitive Collins-Soper angle $\cos\theta^*$ with an analytic expression at the reconstruction level. We conclude our findings in Section 5.

## 2 Symbolic regression

At the core of physics stands the claim that observations can be understood mathematically. For a set of data points from an experimental measurement, there has to be a function $f(x)$ that describes the distribution of those points. In some cases, the approximate form of $f(x)$ is known. For example, decaying particles are described by

$$N(t) = a \cdot e^{-bt} + c . \tag{1}$$

The unknown parameters $a, b, c$ can be inferred from the data.

In the absence of a priori information about $f(x)$ this method fails. On the numerical side, neural networks and on the analytic side symbolic regression (SR) do not assume a specific class of functions. Both determine a general function from a minimization problem

$$\hat{f} = \underset{f \in \mathcal{F}}{\arg\min} \, \mathcal{L}\left(Y, f(\mathbf{X})\right), \tag{2}$$

where $Y \in \mathbb{R}^{N_{\text{data}}}$ ($Y \in [0, 1]^{N_{\text{data}}}$) are the labels in a regression (classification) task with $N_{\text{data}}$ data points, and $\mathbf{X} \in \mathbb{R}^{N_{\text{data}} \times N_f}$ are the inputs with $N_f$ features. For SR, $\mathcal{F}$ is the function space spanned by a set of basic operators defined by the user.

Solving Eq. (2) is a challenging, NP-hard problem [34]. Advanced methods are needed to determine $\hat{f}$ in a reasonable amount of time. SR algorithms have been shown to be able to tackle this problem for example by inferring well-known astrophysical formulas [35–37] and in cosmology [38]. In the following, we compare two different SR-approaches, PySR and SymbolNet, for different tasks.

### 2.1 PySR

PySR [31] is a Python module for symbolic regression that builds on an evolutionary algorithm and performs multiple evolutions at once. The formula finding happens via an inner and an outer loop. The inner loop corresponds to a single evolutionary algorithm with some modifications to speed up the training and improve the performance. The outer loop consists of a training part, where multiple populations evolve independently, and a migration part, in which the populations exchange expressions.

For the evolutionary algorithm defining the inner loop, formulas are represented by trees. Two simple examples can be seen in Fig. 1. The outermost leaves are variables and constants,

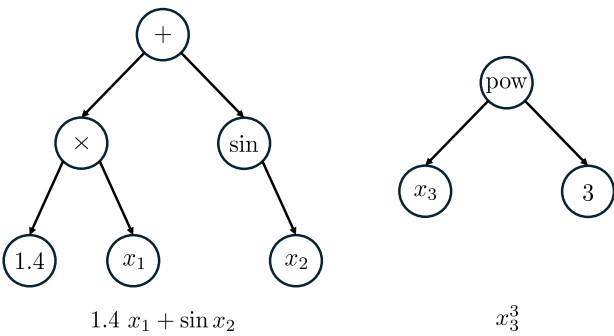

Figure 1: Exemplary function trees displaying the functions $1.4 \, x_1 + \sin x_2$ and $x_3^3$ in PySR.

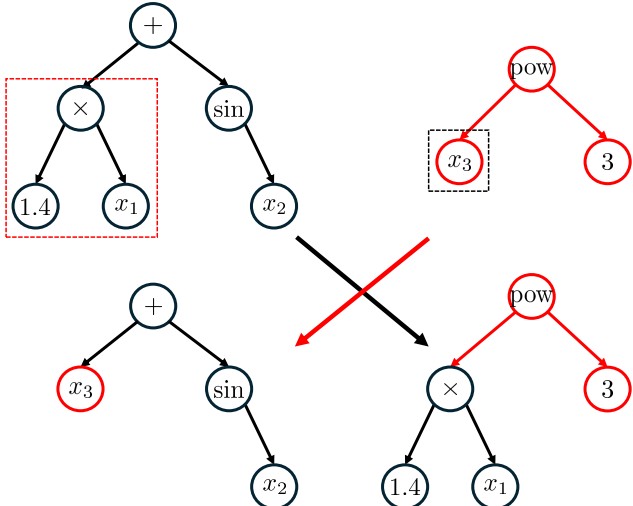

Figure 2: A crossover operation in which the red-dashed part of the original tree has been exchanged with the black-dashed part of another tree. As a result the formulas are modified to $x_3 + \sin x_2$ and $(1.4\, x_1)^3$.

combined by mathematical operations. These consist of unary and binary operations, either modifying a single leaf or combining two leaves, respectively.

A population consists of a number of formula trees. Modifications to a single member of the population happen via tournament selection, where a subset of trees is compared by fitness. The fitness is determined by an objective function acting on the data. A single tree is chosen with a probability proportional to its fitness, to be cloned and modified in the next step. Modifications can include optimization of constants, simplification, mutation of a single node, and a crossover operation. In the latter, two trees exchange part of their structure, as shown in Fig. 2. The new formula tree replaces the one with the lowest fitness in the population.

In contrast to other evolutionary algorithms, PySR can use simulated annealing, where modifications of the trees can be rejected. A mutation is accepted according to the probability

$$p = \exp \frac{L_f - L_i}{\alpha T} \; . \tag{3}$$

Here, $L_f - L_i$ is the fitness difference from the modification, $T \in [0, 1]$ is the temperature, varied during training, and $\alpha$ is a hyperparameter controlling the strength of the annealing. We use a modified version of Eq. (3) [20], where the rejection probability reads

$$p = \exp \frac{L_f - L_i}{\alpha T L_i} \; . \tag{4}$$

This change serves the purpose of not discarding functions with a better shape but lower fitness due to missing constant optimization.

Furthermore, PySR alternates between evolution phases during which trees are primarily mutated and those during which they are primarily optimized. The mutation phase diversifies the population and finds expressions requiring an intermediate state, which would otherwise be simplified. The optimization phase improves the performance of the algorithm. Recently, PySR introduced an adaptive parsimony. The parsimony parameter punishes the complexity of the formulas via a regularization term. Adaptive parsimony instead punishes complexities appearing more often in the population. This forces the algorithm to learn formulas of all

complexities and explore a broader range of functions. Because the evolution of a population is achieved via the inner loop. the training of multiple populations can be parallelized.

In the outer loop, after a fixed number of iterations, the populations communicate and migrate members between them. The overall fittest members of each complexity are stored in the hall of fame. From here, expressions can also migrate to the populations. At the end of the training, the formulas stored in the hall of fame are returned. The final formula is determined by selection criteria such as the highest score evaluated by PySR, or the lowest overall loss.

PySR is powerful and highly customizable, outperforming other algorithms designed for similar problems [31]. Its limitations appear for a large number of input features, or when a formula of high complexity is needed. Here, evolutionary algorithms struggle to converge within a limited time. Furthermore, the tree representation cannot be straightforwardly extended to other structures, like 4-vectors.

## 2.2 SymbolNet

The fact that evolutionary algorithms do not perform well for high-dimensional parameter spaces motivates the use of backpropagation. SymbolNet [32] starts with a fully connected multi-layer perceptron (MLP) and replaces the activation functions with mathematical operators. They either modify a single node or combine nodes. The hidden layers are dubbed symbolic layers. During the so-called sparsity training the mathematical operators and the connections between nodes can be pruned to create a sparsely connected network. This prevents overtraining and simplifies the output formula for improved interpretability.

We change SymbolNet in two ways. First, we adapt it to support 4-vectors by adding a vector dimension to the nodes. All operations are applied to the 4-vectors element-wise, and the vector dimension is collapsed in a specialized symbolic layer. This vectorized SymbolNet requires extra checks because some mathematical operations can turn the 4-vectors unphysical. Second, we divide the training into three steps, to prevent gradient instabilities:

- default training, where only the usual NN weights and biases are trainable;
- mixed training, where every non-auxiliary parameter is trainable [32]; and
- sparsity training, only allowing for pruning of operators and connections.

The core of SymbolNet are two TensorFlow custom layer classes, the input layer and the symbolic layer. They come with trainable and untrainable weights and thresholds, which are summarized in Table 1 and are explained in the following.

The input features $\mathbf{X}_{\text{in}}$ are associated with an auxiliary weight vector and with a threshold vector. The weight vector entries are not trainable and fixed to one. The threshold vector is clipped to the range $[0,1]$ and trainable during sparsity training. The input layer prunes input features which are not relevant for the training via

$$\mathbf{X}_{\text{in}} \rightarrow \mathbf{X}_{\text{in}} \theta (\mathbf{1} - \mathbf{T}_{\text{in}}) . \tag{5}$$

where $\theta$ is the Heaviside step function. Without sparsity training, the input layer simply passes the input to the first symbolic layer.

Symbolic layers are split into two parts. The first one is a linear combination of the incoming features associated with the usual weights and biases of an MLP, complemented by threshold matrices. The default weights are initialized using a random uniform distribution, while the thresholds are initialized to zero. During default training, only the weights and biases are trainable, while in the sparsity setup, only the thresholds are. In the mixed setup, all

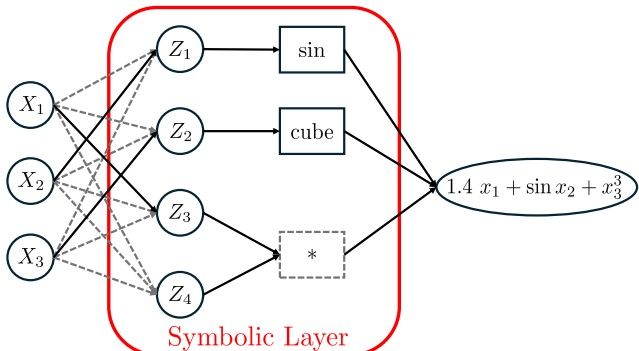

Figure 3: `SymbolNet` architecture with a single symbolic layer. The input parameters are linearly transformed to an input representation for the symbolic layer. The mathematical operations are applied in order, and the output of the symbolic layer is linearly transformed to the final prediction.

four parameters are trainable. Pruning is implemented as

$$w_{ij} \rightarrow w_{ij}\theta(|w_{ij}| - t_{w,ij}),$$
$$b_i \rightarrow b_i\theta(|b_i| - t_{b,i}).$$

(6)

In the second step, the output of the linear operation is passed to a set of pre-defined mathematical operations. This replaces the usual MLP activation. The operations are split into unary and binary operators. Similarly to the input features, each unary (binary) operator is associated with an untrainable auxiliary weight and a threshold, where the latter is trainable

| Layer | Description | Label | Dim. | Values | Trainable | Dim. of trafo |
|---|---|---|---|---|---|---|
| Input layer | Input weights | $\mathbf{W}_{\text{in}}$ | $n_{\text{in}}$ | 1 | × | $n_{\text{in}} \times l$ |
| | Input thresholds | $\mathbf{T}_{\text{in}}$ | $n_{\text{in}}$ | $[0,1]$ | M, S | $\rightarrow n_{\text{in}} \times l$ |
| Linear trafo in symbolic layer | Weights | $w_{ij}$ | $n \times m \times l$ | $\mathbb{R}$ | D, M | |
| | Weight thresholds | $t_{w,ij}$ | $n \times m \times l$ | $\mathbb{R}^+$ | M, S | $n \times l$ |
| | Biases | $b_i$ | $m \times l$ | $\mathbb{R}$ | D, M | $\rightarrow m \times l$ |
| | Bias thresholds | $t_{b,i}$ | $m \times l$ | $\mathbb{R}^+$ | M, S | |
| Symbolic trafo in symbolic layer | Unary weights | $w_{\text{unary},i}$ | $n_f$ | 1 | × | |
| | Unary thresholds | $t_{\text{unary},i}$ | $n_f$ | $[0,1]$ | M, S | $m \times l$ |
| | Binary weights | $w_{\text{binary},i}$ | $n_g$ | 1 | × | $\rightarrow k \times l'$ |
| | Binary thresholds | $t_{\text{binary},i}$ | $n_g$ | $[0,1]$ | M, S | |

Table 1: Parameters used for the training of `SymbolNet` alongside their dimension, possible values of their components, and the phases in which they are trainable. $n_f$ and $n_g$ are the number of unary and binary operators in the layer, while $m = n_f + 2n_g$ and $k = n_f + n_g$. $l, l' = \{1, 4\}$, depends on the type of symbolic layer. D, M, and S stand for the training phases: default, mixed, and sparsity.

during sparsity training. They are pruned via

$$f_i(a) \rightarrow f_i(a)\,\theta(1-t_{\text{unary},i}) + a\left(1-\theta(1-t_{\text{unary},i})\right)$$
$$g_i(a,b) \rightarrow g_i(a,b)\,\theta(1-t_{\text{binary},i}) + (a+b)\left(1-\theta(1-t_{\text{binary},i})\right). \tag{7}$$

Unary operators are simplified to the identity if the threshold parameter exceeds one, while binary operators are simplified to an addition. In the special case of a symbolic layer that takes vectors as its input and returns scalars, the pruning implies that masked operators return zero,

$$f_i(a) \rightarrow f_i(a)\,\theta(1-t_{\text{unary},i})$$
$$g_i(a,b) \rightarrow g_i(a,b)\,\theta(1-t_{\text{binary}},i). \tag{8}$$

For our vectorized `SymbolNet`, we split the symbolic layers into three types, depending on their input and output dimension, and illustrated in Fig. 4.

- $V \rightarrow V$ layers:
  Input and output of the symbolic layer are 4-vector-like objects. All operations are applied element-wise, with the exception of the Lorentz boost

$$g(p_i, p_j) \in \{\text{boost}(p_i|p_j),\dots\}, \tag{9}$$

  which boost the vector $p_i$ to the frame $p_j$ and which is unique to the $V \rightarrow V$ layer. There is no restriction to the number of $V \rightarrow V$ layers in `SymbolNet`.

- $V \rightarrow S$ layer:
  Next, the 4-vector-like objects are transformed to scalars. Possible unary operators are

$$f(p) \in \left\{p_0, p_x, p_y, p_z, \|p\|_3, \|p\|_4\right\}. \tag{10}$$

  The $p_i$ pick a component of the vector, $\|p\|_3$ computes the Euclidean three-vector norm, and $\|p\|_4$ the Minkowskian norm. The binary operators consist of the three-vector and Minkowski products

$$g(p_i, p_j) \in \left\{\langle p_i \times p_j \rangle_3, \langle p_i \times p_j \rangle_4\right\}. \tag{11}$$

  Exactly one $V \rightarrow S$ layer is required in the vectorized `SymbolNet`. All operations in this layer are unique to it.

- $S \rightarrow S$ layers:
  They correspond to the default `SymbolNet` layers and only work with scalar quantities. Any number of these layers may be used.

- Output layer:
  The final symbolic layer in the network is an $S \rightarrow S$ layer that only consists of a linear combination $\mathbb{R}^n \rightarrow \mathbb{R}$ and no further unary or binary operations.

During backpropagation, to ensure stability of the gradient, the step function responsible for masking the operations is replaced by the derivative of the sigmoid function

$$\frac{d\theta(x)}{dx} \simeq \frac{\kappa e^{-\kappa x}}{(1+e^{-\kappa x})^2} \tag{12}$$

with $\kappa = 5$ as used in [32]. The training loss is

$$\mathcal{L} = \mathcal{L}_{\text{base}} + \mathcal{L}_{\text{sparse}}, \tag{13}$$

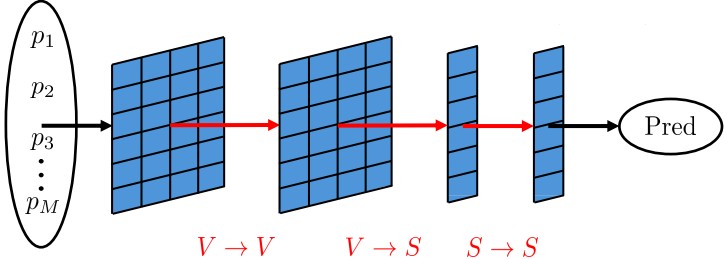

Figure 4: illustration of a vectorized `SymbolNet` structure. The input 4-vectors are transformed via one of each layer types to the final prediction.

where $\mathcal{L}_{\text{base}}$ is the default loss, i.e. Mean Squared Error (MSE) or the cross-entropy. The additional sparsity loss controls the dynamic pruning and is set to zero during the default training. It is given by

$$\mathcal{L}_{\text{sparse}} = \mathcal{L}_{\text{error}} D(s_{\text{weight}}; \alpha_{\text{weight}}, d) \mathcal{L}_{\text{threshold}}^{\text{weight}}$$
$$+ \mathcal{L}_{\text{error}} D(s_{\text{aux}}; \alpha_{\text{aux}}, d) \mathcal{L}_{\text{threshold}}^{\text{aux}} . \tag{14}$$

The target sparsities and the aggressiveness of the pruning are controlled via the decay factor

$$D(s; \alpha, d) = \exp\left[ -\left( \frac{\alpha}{\alpha - \min(s, \alpha)} \right)^d + 1 \right] , \tag{15}$$

where $s$ are the current sparsity values, $\alpha$ the target sparsities set by the user, and $d$ is controlling the aggressiveness of the pruning. We use the default choice $d = 0.01$. To encourage large threshold values, we use

$$\mathcal{L}_{\text{threshold}}^{\text{weight}} = \frac{1}{n_{\text{weight}}} \sum_i e^{-t_{w,i}} \tag{16}$$

$$\mathcal{L}_{\text{threshold}}^{\text{aux}=\{\text{input,unary,binary}\}} = \exp\left( \frac{1}{n_{\text{aux}}} \sum_i^{n_{\text{aux}}} t_{\text{aux},i} \right) . \tag{17}$$

## 3  $\mathcal{CP}$-odd observables for WBF-Higgs production

In order to conclusively test $\mathcal{CP}$ for example in WBF Higgs production, one needs to learn and employ the optimal $\mathcal{CP}$-odd observable for this process. The standard theory framework for this question is the dimension-6 SMEFT operator

$$\frac{c_{H\widetilde{W}}}{\Lambda^2} \varphi^\dagger \varphi \widetilde{W}_{\mu\nu}^I W^{I\mu\nu} , \tag{18}$$

where $\varphi$ is the Higgs doublet, $W$ the $SU(2)_L$ field strength, $\widetilde{W}$ its dual, and $I$ the $SU(2)_L$ index. Two more dimension-6 operators induce $\mathcal{CP}$ violation in WBF Higgs production at the tree level,

$$\varphi^\dagger \varphi \widetilde{B}_{\mu\nu} B^{\mu\nu} \qquad \text{and} \qquad \varphi^\dagger \tau^I \varphi \widetilde{W}_{\mu\nu}^I B^{\mu\nu} . \tag{19}$$

All three operators induce the same Lorentz structure in the $HZZ$ coupling, but only $c_{H\widetilde{W}}$ affects the $HWW$ coupling. This is why we can stick to $c_{H\widetilde{W}}$ for simplicity. In this study, we consider the $H \to \gamma\gamma$ decay channel, taking into account also the effect of $c_{H\widetilde{W}}$ on the decay rate.

### 3.1 Optimal $\mathcal{CP}$-observable

Considering a process with a contribution proportional to the $\mathcal{CP}$-violating coupling $\theta$, we can write the squared amplitude as

$$|\mathcal{M}|^2 = |\mathcal{M}_{\mathcal{CP}\text{-even}}|^2 + 2\theta \operatorname{Re}\left[\mathcal{M}_{\mathcal{CP}\text{-even}}\mathcal{M}_{\mathcal{CP}\text{-odd}}^*\right] + \theta^2|\mathcal{M}_{\mathcal{CP}\text{-odd}}|^2 . \tag{20}$$

The first and last terms are effectively $\mathcal{CP}$-even, while $\mathcal{CP}$ violation only appears through the interference. Correspondingly, we split the single-event likelihood into a $\mathcal{CP}$-even and a $\mathcal{CP}$-odd part,

$$p(x|\theta) = \frac{1}{\sigma(\theta)}\frac{d^d\sigma(x|\theta)}{dx^d} = p_e(x|\theta) + p_o(x|\theta) . \tag{21}$$

Here $\sigma(\theta)$ is the total cross-section and $d^d\sigma/dx^d$ the differential cross section with respect to the observable $x$. The optimal $\mathcal{CP}$-odd observable for a given $\theta$ is [7]

$$\omega_{\mathcal{CP}\text{-odd}} = \frac{p_o}{p_e} . \tag{22}$$

Using ML, a classifier converges to this optimal observable when trained to distinguish two samples with finite values $\pm\theta$. Using Eq.(21) we can generate the two samples drawing from $p_e(x|\theta) \pm p_o(x|\theta)$, so the Neyman-Pearson-optimal classifier is

$$\begin{aligned}
D(x) &= \frac{p_e(x|\theta) + p_o(x|\theta)}{p_e(x|\theta) + p_o(x|\theta) \,+\, p_e(x|\theta) - p_o(x|\theta)} \\
&= \frac{1 + \omega_{\mathcal{CP}\text{-odd}}(x)}{2} \\
\Longleftrightarrow \qquad \omega_{\mathcal{CP}\text{-odd}}(x) &= 2D(x) - 1 .
\end{aligned} \tag{23}$$

This strategy can be applied for any classifier, including boosted decision trees (BDTs) [15].

Using symbolic regression has the advantage that we can check analytically if the learned observable is indeed $\mathcal{CP}$-odd. To define the classification task we assign the label '1' to the sample with positive $c_{H\widetilde{W}}$ and '0' to the sample with negative $c_{H\widetilde{W}}$. The classifier output is mapped to the interval $[0,1]$ as

$$D(x) = \operatorname{sigmoid}(d(x)) , \tag{24}$$

where $d(x)$ is the analytic expression we target with symbolic regression. Since $2\operatorname{sigmoid}(x) - 1$ is odd under $x \to -x$, the $\mathcal{CP}$-properties of $D$ and $\omega_{\mathcal{CP}\text{-odd}}$ are the same.

To improve the training we add an additional term to the cross-entropy loss $\mathcal{L}_{\text{CE}}$, penalizing non-$\mathcal{CP}$-odd observables,

$$\mathcal{L} = \mathcal{L}_{\text{CE}} + \frac{\alpha}{n_{\mathcal{CP}}}\sum_{i=1}^{n_{\mathcal{CP}}}\left[d(x_i) + d(x_i^{\mathcal{CP}})\right] , \tag{25}$$

where $x_i^{\mathcal{CP}}$ is the $\mathcal{CP}$-transformed input. The sum runs over $n_{\mathcal{CP}}$ phase space points and $\alpha$ balances the two loss contributions. If $d$ is $\mathcal{CP}$-odd, then $d(x_i^{\mathcal{CP}}) = -d(x_i)$ and $\mathcal{L} = \mathcal{L}_{\text{CE}}$.

At this point a difference between `PySR` and `SymbolNet` becomes relevant: `PySR` starts with simple expressions and generates more complex expressions over time. The additional $\mathcal{CP}$-odd loss of Eq.(25) prevents the generation of non-$\mathcal{CP}$-odd formulas. While this prevents $\mathcal{CP}$-even equations as intermediary mutation steps, we find this to not affect performance. In contrast, `SymbolNet` starts from a complicated expression and prunes it over time. The $\mathcal{CP}$-odd loss term does not necessarily result in $\mathcal{CP}$-odd formulas.

### 3.2 Events and training

We generate leading-order events for the hard process

$$pp \to H_{\gamma\gamma} jj \qquad (26)$$

using MADGRAPH5_AMC@NLO 3.5.0 [39] with the SMEFTSIM UFO model [40, 41]. For the parton shower, we use PYTHIA8 [42]; for the detector simulation, DELPHES [43]; and for the jet clustering, FASTJET [44]. We generate data for 11 different scenarios,

$$c_{H\widetilde{W}} \in \{0, \pm 0.1, \pm 0.25, \pm 0.5, \pm 0.75, \pm 1\} . \qquad (27)$$

Unless mentioned otherwise, we use 250k events for training and testing as well as 100k events for validation. For the analysis we require exactly two photons and at least two tagging jets and impose the pre-selection cuts

$$m_{\gamma\gamma} = 110 \dots 140 \text{ GeV} , \qquad \frac{p_{T,\gamma_{1,2}}}{m_{\gamma\gamma}} > 0.35, 0.25 ,$$

$$p_{T,j} > 30 \text{ GeV} , \qquad |\eta_j| < 4.4 , \qquad \Delta\eta_{jj} > 2 . \qquad (28)$$

The Higgs 4-momentum is reconstructed from the two photons.

Following Sec. 3.1 we train our WBF $\mathcal{CP}$-odd observables through a classifier to distinguish events with positive and negative $c_{H\widetilde{W}}$. We apply a sigmoid function to the learned formulas to ensure $D \in [0, 1]$. In our training, we use

$$\left\{ \quad x_{j_{1,2}} = \frac{p_{T,j_{1,2}}}{m_h}, \eta_{j_1}, \phi_{j_1}, \quad \Delta\eta_{jj}, \Delta\phi_{jj}, x_{jj} = \frac{m_{jj}}{m_h}, \quad x_h = \frac{p_{T,h}}{m_h}, \eta_h, \phi_h \quad \right\} \qquad (29)$$

for jets ordered in $p_T$ and with $m_h = 125$ GeV.

For PySR, we train for 500 iterations with a maximum complexity of 60 using sine, cosine, absolute value, exponential, logarithm, sine hyperbolicus, cosine hyperbolicus, addition, multiplication, and division as operators. After training, we optimize the numerical constants. As our `SymbolNet` network, we use two symbolic layers with the functions sine, cosine, absolute value, square, square root, multiplication, and division. We here use non-vectorized symbolic layers since the number of input features is comparably small. We train the network for 1000 epochs with $\alpha_{\text{input}} = 0.6$, $\alpha_{\text{weight}} = 0.6$, $\alpha_{\text{unary}} = \alpha_{\text{binary}} = 0.3$. No afterburning is necessary, since the loss minimization already optimizes the numerical constants.

### 3.3 Results and formulas

In addition to the $\omega_{\mathcal{CP}\text{-odd}}$ distributions, we also compute the bin-wise asymmetry

$$\mathcal{A}_i = \frac{N_i^+ - N_i^-}{N_i^+ + N_i^-} \qquad \text{with} \qquad N_i^+ = \#[\omega_{\mathcal{CP}\text{-odd},i}, \omega_{\mathcal{CP}\text{-odd},i+1}]$$

$$N_i^- = \#[-\omega_{\mathcal{CP}\text{-odd},i+1}, -\omega_{\mathcal{CP}\text{-odd},i}] , \qquad (30)$$

where $\#[a, b]$ denotes the number of events in the interval from $a$ to $b$. For the statistical uncertainty of the $\mathcal{A}_i$ we use an integrated luminosity of $300 \, \text{fb}^{-1}$. Details on the calculation of the standard deviations can be found in App. B.

In the left panel of Fig. 5 we show the observable

$$\frac{1}{m_h^2} p_{T,j_1} p_{T,j_2} \sin \Delta\phi_{jj} , \qquad (31)$$

the most sensitive $\mathcal{CP}$-odd observable at the parton level for small $c_{H\widetilde{W}}$ [11, 20, 45, 46]. It provides a baseline for our symbolic regression. While the SM distribution is symmetric around zero, the distributions with $\mathcal{CP}$ violation are asymmetric.

The relation between $\mathcal{CP}$ and this bin-wise asymmetry is confirmed in the right panel of Fig. 5. The SM distribution is symmetric, with small statistical fluctuations. For $c_{H\widetilde{W}} = 1$ we see a positive asymmetry, while for $c_{H\widetilde{W}} = -1$ the asymmetry also changes sign. The higher bins feature a larger asymmetry in comparison to the inner bins, but with a larger statistical uncertainty, suggesting that there will be a sweet spot for the analysis.

Our learned formulas from PySR and SymbolNet are evaluated in the left panels of Fig. 6. Both are very similar, the SymbolNet distribution being slightly wider than its PySR counterpart. Both reflect the $\mathcal{CP}$-odd nature, with a symmetric SM distribution and the asymmetric but mirrored outcomes for $c_{H\widetilde{W}} = \pm 1$. Compared to $p_{T,j_1} p_{T,j_2} \sin \Delta \phi_{jj}$ in Fig. 5, the PySR and SymbolNet distributions are wider and have less separation around zero. The analysis power of the learned formulas is illustrated in the right panels of Fig. 6. Both formulas have the largest asymmetry for the highest bins, while the most significant bins are in the middle.

Finally, we can confirm the $\mathcal{CP}$-odd nature of the learned formulas from the analytic forms,

$$d^{\texttt{PySR}} = \frac{1.8566 \sin \Delta \phi_{jj}}{\left| \dfrac{0.3080 x_{j_1} \log \Delta \eta_{jj} + \log \Delta \eta_{jj} \sinh(x_{j_2} - 2.5977) + 0.3080 \sinh x_h}{x_{j_1} \log \Delta \eta_{jj} + \sinh x_h} \right| + 0.6047}$$

$$d^{\texttt{SymbolNet}} = 0.715 \Delta \phi_{jj} \Big[ -0.348(0.561 x_{j_2} - 0.25 \Delta \eta_{jj} + 0.0315 x_{jj} + 0.746 x_h) - 0.27 \Big]$$
$$\cdot \Big[ 0.0493(0.603 \Delta \eta_{jj} - 0.0811 x_{jj} - x_h)^2$$
$$- 0.654 |0.463 x_{j_2} + 0.477 \Delta \eta_{jj} + 0.373 x_h|^{0.5}$$
$$- 0.134 \sin(-0.555 \Delta \eta_{jj} + 0.345 x_{jj} + 0.443 x_h)$$
$$- 1.82 \cos(0.642 \Delta \phi_{jj}) \Big] , \tag{32}$$

where we color the $\mathcal{CP}$-odd part, to confirm that the learned observables are actually $\mathcal{CP}$-odd. We observe that a simple $\mathcal{CP}$-odd structure is multiplied by a more complex $\mathcal{CP}$-even function, retaining the $\mathcal{CP}$-odd nature of the overall expression. While for PySR, we explicitly enforce this by including the $\mathcal{CP}$-odd loss contribution, we do not include it in the SymbolNet training. As we will see below, a $\mathcal{CP}$-odd loss destabilizes the training. By construction, the

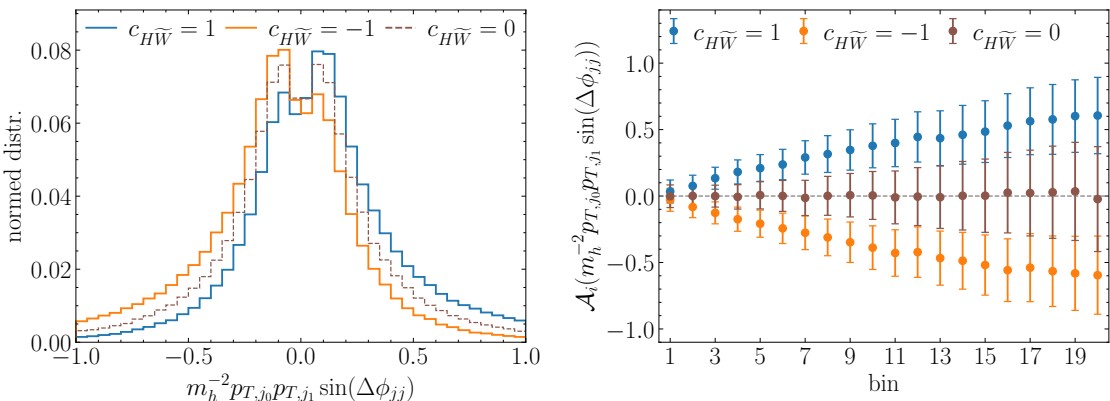

Figure 5: Normalized distribution of $p_{T,j_1} p_{T,j_2} \sin \Delta \phi_{jj}$ (left) and its bin-wise asymmetry $\mathcal{A}$ defined in Eq. 30 (right) for the SM (corresponding to $c_{H\widetilde{W}} = 0$) and $c_{H\widetilde{W}} = \pm 1$.

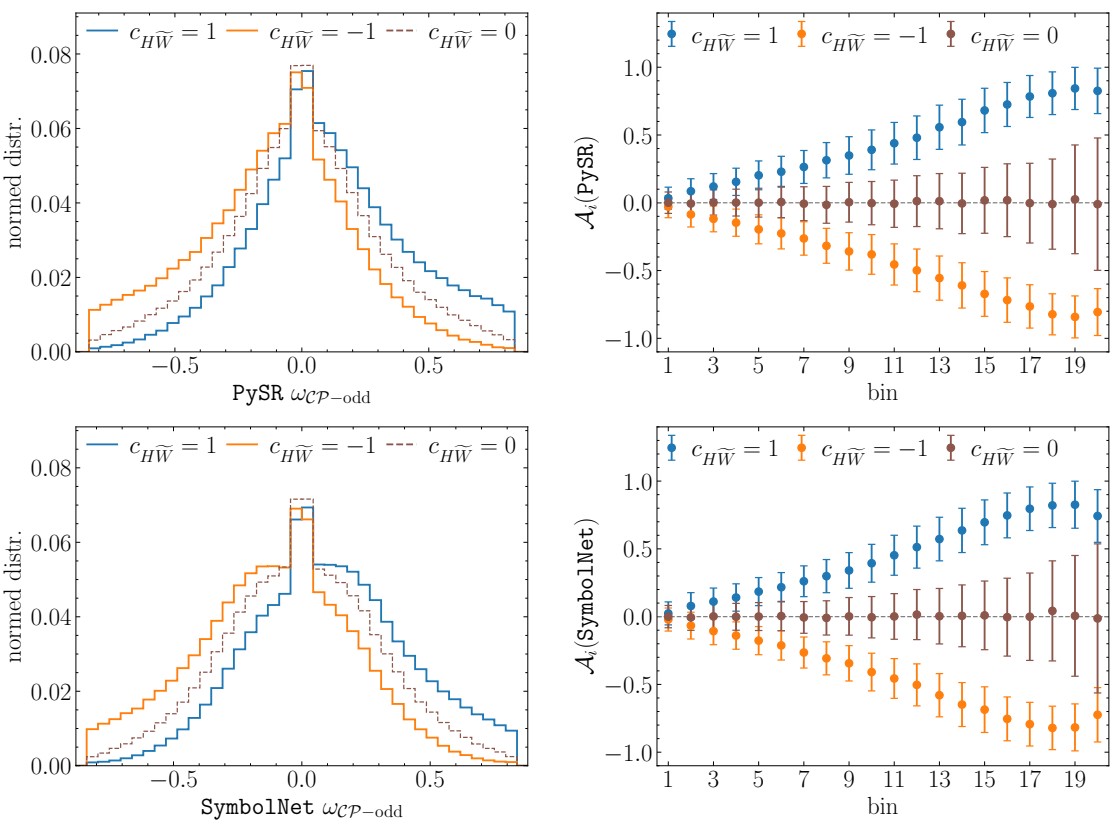

Figure 6: Performance of the learned formulas for the $\mathcal{CP}$-odd observable from PySR and SymbolNet (left) and their bin-wise asymmetries $\mathcal{A}$ (right).

learned SymbolNet formula does not have to be $\mathcal{CP}$-odd, and the formula learned in another SymbolNet run — using the same settings but a different random initialisation — confirms this,

$$
\begin{aligned}
d^{\texttt{SymbolNet}} = & -0.19\Big[0.87\Delta\phi_{jj} + 0.063x_h\Big]\Big[-0.66x_{j_2} + 0.050\Delta\phi_{jj} - 1.1x_h - 0.02\phi_h\Big] \\
& -0.65\sin\Big[0.49x_{j_2} + 0.23\Delta\phi_{jj} + 0.28x_h \\
& \qquad -(0.082\Delta\phi_{jj} + 0.0059x_h)(0.66x_{j_2} - 0.05\Delta\phi_{jj} + 1.1x_h + 0.02\phi_h) \\
& \qquad +(0.35\Delta\phi_{jj} + 0.18x_h)(-0.60x_{j_2} - 0.66\Delta\phi_{jj} - 0.36x_h) \\
& \qquad +0.42\cos(0.50x_{j_2} - 1.3\Delta\phi_{jj} + 0.46x_h)\Big] \\
& +0.62\cos\Big[0.50x_{j_2} - 1.3\Delta\phi_{jj} + 0.46x_h\Big].
\end{aligned}
\tag{33}
$$

Even though it discriminates the SM data better from the $c_{H\widetilde{W}} = \pm 1$ cases, it is not $\mathcal{CP}$-odd, so a deviation from the SM based on this observable is not a unique sign of $\mathcal{CP}$ violation.

From the bin-wise asymmetry we can calculate significances for 300fb$^{-1}$, as detailed in App. B. Since we are constraining only one degree of freedom, the significance is effectively given by the square root of the $\chi^2$ value. These significances are listed in Tab. 2. The significances for negative $c_{H\widetilde{W}}$ are identical to their positive counterparts. For comparison, we also show significances from a numerical BDT, trained using XGBoost [47, 48], and SymbolNet without pruning any network components, labelled as "SymbolNet full". As we know from our two SymbolNet formulas, the BDT and full SymbolNet observables can have $\mathcal{CP}$-even components enhancing the signal significance. However, this does not imply a higher signifi-

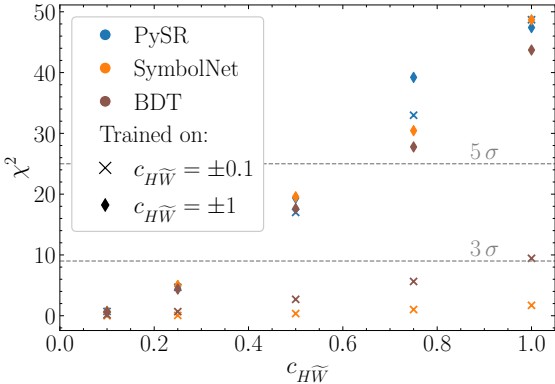

Figure 7: Dependence of the test significance on $c_{H\widetilde{W}}$. For the diamonds, the observables are trained with $c_{H\widetilde{W}} = \pm 1$, while for the crosses the observables are trained with $c_{H\widetilde{W}} = \pm 0.1$. For the two lowest $c_{H\widetilde{W}}$ values, the blue crosses are hidden behind the diamond markers.

cance of discovering $\mathcal{CP}$ violation. While we can check this artifact for the formulas from PySR and SymbolNet, this is not possible for a BDT or neural-network classifier.

The significances quoted for $c_{H\widetilde{W}} = 1$ vs SM lie around $7\,\sigma$ for the case where the training data is with $c_{H\widetilde{W}} = \pm 1$. The learned SymbolNet and PySR formulas slightly outperform the classic $p_{T,j_1}p_{T,j_2}\sin\Delta\phi_{jj}$. Training on $c_{H\widetilde{W}} = \pm 0.1$, i.e. with a much smaller $\mathcal{CP}$-odd contribution, the performance of the SymbolNet and BDT observables drops. In contrast, the PySR formula is robust to the less sensitive training data. For a weaker signal, $c_{H\widetilde{W}} = 0.25$, but still trained on $c_{H\widetilde{W}} = \pm 0.1$, the typical significances shrink to around $2.5\,\sigma$, but the performance pattern of the different approaches remains, i.e. PySR performs better than the BDT and SymbolNet.

In Fig. 7 we investigate the dependence of the performance of the learned formulas on the size of $c_{H\widetilde{W}}$ in the training data. It confirms the pattern observed in Tab. 2 — since PySR builds formulas from simple to complicated, it is very efficient even for little $\mathcal{CP}$ violation in the training data. The PySR $\chi^2$ values are stable for formulas trained on $c_{H\widetilde{W}} = \pm 0.1 \dots \pm 1$. In contrast, the SymbolNet and BDT performances suffer for less $\mathcal{CP}$ violation in the training

| | $\sigma(c_{H\widetilde{W}} = 1$ vs. SM$)$ | $\sigma(c_{H\widetilde{W}} = 0.25$ vs. SM$)$ |
|---|---|---|
| $p_{T,j_1}p_{T,j_2}\sin\Delta\phi_{jj}$ | 6.76 | 2.43 |
| trained on $c_{H\widetilde{W}} = \pm 1$ | | |
| PySR | 6.98 | 2.47 |
| SymbolNet | 7.07 | 2.49 |
| SymbolNet full | **7.84** | **2.51** |
| BDT | 6.71 | 2.36 |
| trained on $c_{H\widetilde{W}} = \pm 0.1$ | | |
| PySR | **7.07** | **2.43** |
| SymbolNet | 1.67 | 0.82 |
| SymbolNet full | 6.18 | 2.12 |
| BDT | 3.27 | 1.26 |

Table 2: Significances for distinguishing the dimension-6 hypothesis from the SM, evaluated using the various learned $\mathcal{CP}$-odd observables. The highest significance in each category is marked in bold.

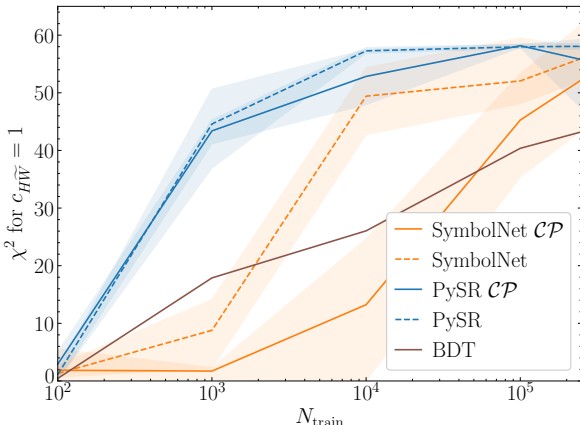

Figure 8: Test significance as a function of the training statistics. The label $\mathcal{CP}$ indicates the additional $\mathcal{CP}$-odd loss of Eq.(25). The curves are averaged over four runs with the orange and blue bands indicating the minimum and maximum performance.

data.

The efficiency of PySR is also helpful if the amount of training data is reduced. This is shown in Fig. 8, where we vary the amount of training data with $c_{H\widetilde{W}} = \pm 1$ and show the significance for distinguishing $c_{H\widetilde{W}} = 1$ from the SM. We show two curves for which the observables are learned with and without punishing non-$\mathcal{CP}$-odd formulas. The learned PySR formulas offer good performance for as little as 1000 training events. Punishing non-$\mathcal{CP}$-odd formulas during the training does affect the performance, but it ensures that the learned formulas are actually $\mathcal{CP}$-odd.

SymbolNet requires ten times more events than PySR to reach a performance plateau. Punishing $\mathcal{CP}$-odd formulas decreases the SymbolNet performance significantly, because the algorithm starts with a complicated and not $\mathcal{CP}$-odd formula. The $\mathcal{CP}$-odd loss then dominates the combined loss, which can only be effectively reduced by deactivating elements of the network early on. These elements, however, can be important for the actual differentiation between the $c_{H\widetilde{W}} = \pm 1$ samples, worsening the performance if including the $\mathcal{CP}$-odd loss contribution. Finally, the BDT does not reach the data efficiency of PySR, either.

# 4  Collin-Soper angle for $t\bar{t}H$ production

Top-associated Higgs production allows to test for $\mathcal{CP}$ violation through the dimension-4 coupling [49]

$$\mathscr{L}_{\text{Yuk}} = -\frac{y_t}{\sqrt{2}}\bar{t}\left(c_t + i\gamma_5 \tilde{c}_t\right)tH \,, \tag{34}$$

where $c_t$ and $\tilde{c}_t$ describe the $\mathcal{CP}$-even and $\mathcal{CP}$-odd contributions to the coupling, respectively. $t$ is the top-quark field; and $H$, the physical Higgs field.

The corresponding $\mathcal{CP}$ angle is

$$\tan\alpha_t = \frac{\tilde{c}_t}{c_t} \,, \tag{35}$$

and a value of 45° approximately corresponds to the current experimental limits [50–53].

This modification of the SM top-Yukawa interaction, which is recovered by setting $c_t = 1$ and $\tilde{c}_t = 0$, is generated by the dimension-6 SMEFT operator

$$\frac{c_{t\phi}}{\Lambda^2}(\varphi^\dagger\varphi)(\bar{Q}_3 t_R \widetilde{\varphi}), \qquad (36)$$

where $Q_3$ is the third-generation quark-doublet and $t_R$ the right-handed top-quark field. The relation between $c_{t\phi}$ and $c_t/\tilde{c}_t$ can be found e.g. in Ref. [54]. Besides the modification of the Higgs–top interaction above, this operator also introduces a Higgs–Higgs–top interaction, which is, however, irrelevant for $t\bar{t}H$ production.

The top-Yukawa interaction of Eq.(34) gives rise to three terms in the squared matrix element, just like in Eq.(20),

$$|\mathcal{M}_{t\bar{t}H}|^2 = c_t^2 \, |\mathcal{M}_{\mathcal{CP}\text{-even}}|^2 + 2c_t\tilde{c}_t \, \text{Re}\left[\mathcal{M}_{\mathcal{CP}\text{-even}}\mathcal{M}_{\mathcal{CP}\text{-odd}}^*\right] + \tilde{c}_t^2 |\mathcal{M}_{\mathcal{CP}\text{-odd}}|^2 \qquad (37)$$

The terms proportional to the squared coupling modifiers are $\mathcal{CP}$-even, while the interference term is $\mathcal{CP}$-odd.

At the LHC, $\mathcal{CP}$ information about the top-Yukawa coupling is primarily obtained from $\mathcal{CP}$-even, but $\mathcal{CP}$-sensitive, observables. $\mathcal{CP}$-odd observables probing the interference term are numerically suppressed [55, 56]. We also show in App. A that learning optimal $\mathcal{CP}$-odd observables does not boost the sensitivity to an observable level.

A powerful $\mathcal{CP}$-sensitive observable is the Collins-Soper (CS) angle [56–60]

$$\cos\theta^* = \frac{\vec{p}_t \cdot \vec{n}}{\|\vec{p}_t\| \cdot \|\vec{n}\|} \qquad (38)$$

which is the angle between the $t\bar{t}$ system and the beam axis $n$ in the $t\bar{t}$ rest frame. Measuring it requires a reconstruction of the $t\bar{t}$ system and is therefore challenging. We train symbolic regression models to reconstruct $\theta^*$ from the final-state momenta, so we can assess $\mathcal{CP}$ information without explicitly reconstructing both top quarks.

## 4.1 Events and training

We use MADGRAPH5_AMC@NLO 3.5.4 [39] to separately generate leading order events for

$$pp \to t\bar{t}H \qquad \text{and} \qquad pp \to t\bar{t}H + j \qquad (39)$$

production. The events are scaled to the NLO rate via a constant $K$-factor of 1.13 [61]. $\mathcal{CP}$ violation in the complex top-Yukawa coupling is introduced via the HC_NLO_X0 UFO model [49]. The semi-leptonic top decays are simulated with MadSpin [62]. The events are at parton level, but we will discuss approximate detector effects later. The acceptance cuts are minimal [63],

$$p_{T,j} > 15\,\text{GeV} \qquad \text{and} \qquad |\eta_{j,\ell}| < 4\,. \qquad (40)$$

The PySR and SymbolNet training datasets contain the 4-momenta of each final state particle, normalized by the top mass, to obtain dimensionless quantities of order one. PySR takes individual 4-vector components as input, while SymbolNet takes the entire 4-vector. Since we decay the tops semi-leptonically, we assume that the $b$-jets have been correctly assigned to the lepton and the light quarks.

To assess the performance in reconstructing $\theta^*$, we define six benchmark scenarios for the information given to PySR and SymbolNet, listed in Tab. 3. The first two scenarios include the full kinematic information at parton level, but in different rest frames. From scenario 3 on, the full neutrino momentum is replaced by $E_T^{\text{miss}}$. Scenarios 4 and 6 include an additional

hard jet, while scenarios 5 and 6 approximate limited detector resolution via smearing. For scenarios 1 and 2, where the full kinematic information is available, we use an MSE loss as $\mathcal{L}_{\text{base}}$. Otherwise, we use an inverse Gaussian loss

$$\mathcal{L}_{\text{InvGaussian}} = 1 - \exp\left[-\frac{(y - \hat{y})^2}{2\left(\dfrac{\max(y)}{\sigma} - \dfrac{\min(y)}{\sigma}\right)^2}\right]. \tag{41}$$

with $\sigma = 8$, which is more robust against outliers. Details on this loss function are provided in Appendix C.

For the training of PySR, the hyperparameters are slightly varied depending on the scenario. In scenario 1, we train for 200 iterations and allow a maximum complexity of 60. The functions given to PySR are the square, square root, addition, and division. For the other scenarios, we expand the functions to also include the sine, cosine, sine hyperbolicus, subtraction, and multiplication. Furthermore, the training iterations are raised to 2000 (3000) and the maxsize to 50 (60) in scenarios 2 to 4 (5 and 6).

Ensuring a stable training of SymbolNet is non-trivial, because the activation functions are replaced with various mathematical operations. Furthermore, the sparsity training can quickly lead to gradient instabilities due to the sudden jumps in the loss when a threshold is reached, pruning a weight or an operator. To improve the training, we split it into the three stages introduced in Sec. 2.2: First, we train with disabled sparsity thresholds, corresponding to a default MLP training. We found that a small initial weight initialization with a uniform distribution over $[-0.02, 0.02]$ gives the most stable results.

After this training, the network with the lowest validation loss is passed to the mixed training, where weights and sparsity thresholds are both trainable parameters. Finally, the weights are disabled and the sparsity thresholds are trained alone. During the latter two training phases, the loss and gradient are monitored, and training is stopped when one of them diverges. We perform hyperparameter scans for each scenario, varying the learning rate, batch size and model complexity. The hyperparameters can be found in Table 4. For optimization, the Adam optimizer was used. The LookaheadAdam optimizer was considered, but did not lead to improved results. The functions used in each layer can be found in Table 5.

| Scenario | Frame | $\nu$ Info | QCD Radiation | Smearing |
|:---:|:---:|:---:|:---:|:---:|
| 1 | $t\bar{t}$ | Full | × | × |
| 2 | Lab | Full | × | × |
| 3 | Lab | $E_T^{\text{miss}}$ | × | × |
| 4 | Lab | $E_T^{\text{miss}}$ | ✓ | × |
| 5 | Lab | $E_T^{\text{miss}}$ | × | ✓ |
| 6 | Lab | $E_T^{\text{miss}}$ | ✓ | ✓ |

Table 3: Benchmark scenarios to assess the performance of PySR and SymbolNet. The scenarios become increasingly more complex.

| Scenario | $N_{\text{epochs}}^{\text{default}}$ | $N_{\text{epochs}}^{\text{mixed}}$ | $N_{\text{epochs}}^{\text{sparsity}}$ | $LR^{\text{default}}$ | $LR^{\text{mixed}}$ | $LR^{\text{sparsity}}$ | $BS$ |
|---|---|---|---|---|---|---|---|
| 1 | 200 | 100 | 20 | 0.01 | 0.005 | 0.001 | 128 |
| 2 | 500 | 100 | 20 | 0.01 | 0.005 | 0.001 | 128 |
| 3 & 4 | 1000 | 400 | 50 | 0.01 | 0.002 | 0.001 | 64 |
| 5 & 6 | 2000 | 500 | 50 | 0.02 | 0.002 | 0.001 | 64 |

Table 4: Hyperparameters for the training of `SymbolNet` in the various scenarios.

## 4.2 Results

The `PySR` and `SymbolNet` predictions for the six scenarios are depicted in Fig. 9. In the simplest scenario 1, the SR algorithms only need to build the top quarks from the provided decay products and learn the angle with respect to the beam axis. Both algorithms perform well and find the exact structure of the analytic formula,

$$\cos\theta^*_{\text{PySR}} = \frac{p_{z,b} + p_{z,\bar{l}} + p_{z,\nu}}{\sqrt{\left(p_{x,b} + p_{x,\bar{l}} + p_{x,\nu}\right)^2 + \left(p_{y,b} + p_{y,\bar{l}} + p_{y,\nu}\right)^2 + \left(p_{z,b} + p_{z,\bar{l}} + p_{z,\nu}\right)^2}} \,,$$

$$\cos\theta^*_{\text{SymbolNet}} = \frac{1.006 p_{z,b} + 1.001 p_{z,\bar{l}} + 1.002 p_{z,\nu} - 1.027 p_{z,\bar{b}} - 1.027 p_{z,q} - 1.031 p_{z,\bar{q}}}{\left\| 1.034 p_b + 1.022 p_{\bar{l}} + 1.024 p_\nu - p_{\bar{b}} - 1.007 p_q - 1.009 p_{\bar{q}} \right\|_3}$$

$$\approx \frac{1}{2}\left( \frac{p_{z,t}}{\|p_t\|_3} - \frac{p_{z,\bar{t}}}{\|p_{\bar{t}}\|_3} \right) , \tag{42}$$

where all variables are defined in the $t\bar{t}$ frame. The only difference is that `SymbolNet` does not tune all weights to exactly one. We traced this back to numerical noise in MADSPIN. Moreover,

| Scenario | Operators | $V \to V$ layer | $V \to S$ layer | $S \to S$ layer | $S \to S$ layer |
|---|---|---|---|---|---|
| 1 | $f(p)$ | – | $\{p_z, \|p\|_3\}$ | $\{id.\}$ | – |
|  | $g(p_i, p_j)$ | – | $\{\langle p_i \times p_j \rangle_3\}$ | $\{/\}$ | – |
| 2 | $f(p)$ | $\{id.\}$ | $\{p_z, \|p\|_3\}$ | $\{id.\}$ | – |
|  | $g(p_i, p_j)$ | $\{boost\}$ | $\{\langle p_i \times p_j \rangle_3\}$ | $\{/\}$ | – |
| 3 & 4 | $f(p)$ | $\{tanh\}$ | $\{p_z, \|p\|_3\}$ | $\{\,{}^2, \sqrt{\,}, \sin, \cos\}$ | – |
|  | $g(p_i, p_j)$ | $\{boost\}$ | $\{\langle p_i \times p_j \rangle_3\}$ | $\{*, /\}$ | – |
| 5 & 6 | $f(p)$ | $\{tanh\}$ | $\{p_0, p_x, p_y, p_z, \|p\|_3\}$ | $\{\,{}^2, \sqrt{\,}\}$ | $\{\sin, \cos\}$ |
|  | $g(p_i, p_j)$ | $\{boost\}$ | $\{\langle p_i \times p_j \rangle_3, \langle p_i \times p_j \rangle_4\}$ | $\{*, /\}$ | $\{+, -\}$ |

Table 5: Types of layers and functions used for the training of `SymbolNet` in the various scenarios.

SymbolNet does not exclude one side of the top decay. Instead, it builds the CS angle twice — once from the leptonic and once from the hadronic top decay. The resulting numerical differences to the true formula are negligible. For the other scenarios, the learned formulas are given in Appendix D. In scenario 2 the variables are given in the lab frame. SymbolNet reaches a similar accuracy as before, since it can apply a boost to the input variables before building the formula. In contrast, PySR cannot mimic a boost from the scalar input variables and shows slight deviations from the truth.

In all other scenarios the missing neutrino information implies that there is no analytic reference formula. Also, since the reconstruction tasks get more and more difficult, the goodness of the fit decreases with increasingly realistic scenarios. In scenarios 3 and 4, where no smearing is applied, PySR and SymbolNet yield very similar results. When the smearing is added for the results in the lower row of Fig. 9, SymbolNet performs better than the simple PySR formula

$$
\cos\theta^*_{\mathrm{PySR}} = \sin\Bigg[\bigg(1.114 p_{z,b} + 2.143 p_{z,\bar{l}} - 0.858 p_{z,\bar{b}} - 0.426 p_{z,q} - 1.088 p_{z,\bar{q}}\bigg)\Bigg/
$$
$$
\bigg(E_T^{\mathrm{miss}} E_q + E_b + E_{\bar{b}} + 1.85 E_{\bar{l}} + E_{\bar{q}}
$$
$$
- \big(p_{x,b} + p_{x,\bar{l}}\big)\big(p_{x,\bar{b}} + p_{x,q} + p_{x,\bar{q}}\big) - 0.863 - \frac{0.205 p_{z,q}}{\sqrt{E_q}}\bigg)\Bigg]. \tag{43}
$$

It yields more accurate predictions for all values of $|\cos\theta^*|$ and consequently also has a lower MSE loss, as detailed below.

An interesting observation in Fig. 9 is that both SymbolNet and PySR accurately fit the regime around $\cos\theta^* \approx 0$, but underestimate the extreme bins with $|\theta^*| \approx 1$. This holds true whenever there is no true analytic formula to be found. A possible explanation is offered by the cyclic property of the CS angle. Any angle outside $\cos\theta^* \in [-1, 1]$ is automatically mapped back into the interval, since it corresponds to the same physical state. This behavior has to be learned via restricting the output to this range, which does not always happen exactly. In some scenarios, SymbolNet predicts values outside of this range, as indicated by the overflow bins in Fig. 9. For PySR, the final output functions are often restricted to a smaller range, e.g. $\cos\theta^* \in [-0.95, 0.95]$, which might explain the underestimation in the outer bins. We found that neither a cyclic loss nor a fixed mapping of the output layer to the range $[-1, 1]$ improve the results.

**Structure of learned formulas**    Although the final formulas shown above and in Appendix D are not very intuitive, there exist some general patterns. First, PySR formulas are much less complex, as it starts from a very simple formula and then increases the complexity. Despite having to predict $\cos\theta^*$, PySR always chooses the core of the formulas as

$$
\cos\theta^*_{\mathrm{PySR}} \sim \sin\frac{\sum_i a_i\, p_{i,z}}{\sum_i b_i\, E_i}\,. \tag{44}
$$

For low complexity, PySR usually uses the energy as a replacement for the 3-vector norm. Similarly, the SymbolNet formulas have a universal structure, but it is more complex,

$$
\cos\theta^*_{\mathrm{SymbolNet}} \sim \frac{\mathrm{boost}\Big(\sum_i a_i\, p_i \mid \sum_j b_j\, p_j\Big)\Big|_z}{\left\|\mathrm{boost}\Big(\sum_i a_i\, p_i \mid \sum_j b_j\, p_j\Big)\right\|_3}\,. \tag{45}
$$

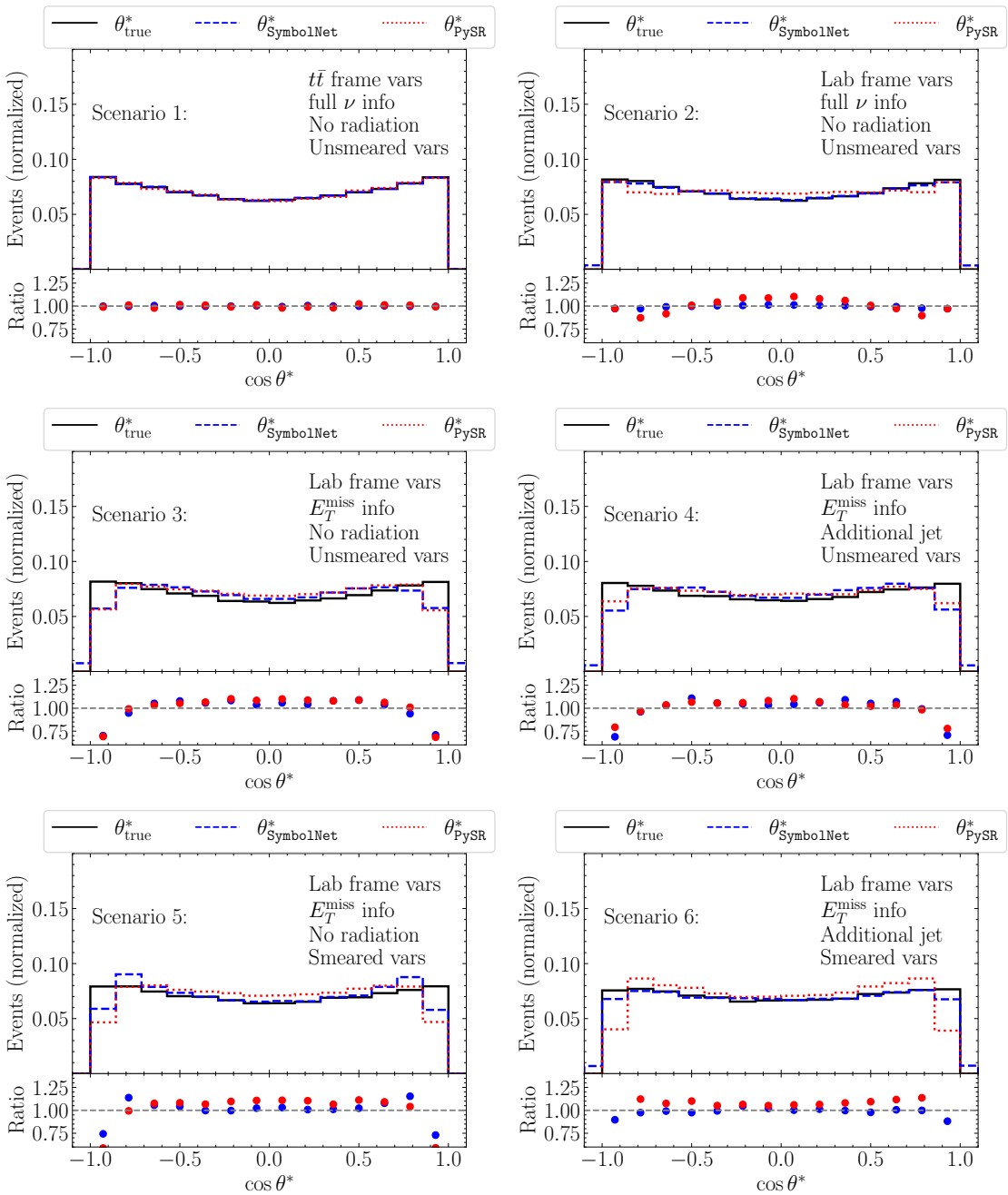

Figure 9: Predicted distributions of the CS angle for the six scenarios defined in Tab. 3. Two overflow bins are added to each distribution.

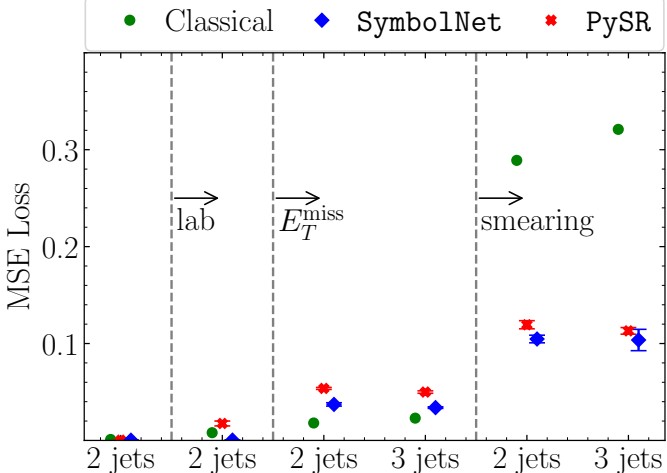

Figure 10: MSE values for each scenario. The values are shown for the classical reconstruction (green), `PySR` (red), and vectorized `SymbolNet` (blue). The central values and error bars correspond to the mean and the standard deviation of ten independent training runs.

This formula reduces to the exact CS angle after including the full neutrino information and for $a_i = b_i = 1$. The two general forms indicate that the structure of the parton-level formulas is also useful at the reco level.

In Fig. 10, we show the MSE values for the `PySR` (red) and `SymbolNet` (blue) formulas evaluated on the test dataset. For each of the six benchmarks both algorithms are trained ten times, and we show the mean and standard deviation of the computed MSE. Runs that do not converge are discarded. The MSE is not necessarily the training objective, instead we use it to assess the goodness of the fit. The MSE values of the two formulas are contrasted with a classical reconstruction of the top quarks [56], where the longitudinal neutrino momentum is reconstructed using the $W$-mass constraint. After that, the light jets, $b$-jets and $W$-boson are combined to two top quarks. We see that this classical reconstruction leads to slightly better MSE values when the neutrino information is missing, but the variables are not smeared. Including detector smearing, the classical reconstruction is clearly outperformed by `PySR` and by `SymbolNet`. This is likely a consequence of the classical reconstruction not respecting the resolution of different objects, while the SR algorithms can account for this via tuning of the parameters. Among the two learned formulas, `SymbolNet` leads to slightly lower MSE values than `PySR`.

**Observable performance**  In Fig. 11, we compare `PySR` and `SymbolNet` for scenario 6, with QCD radiation and detector smearing. The upper panels show the distribution of $\cos\theta^*$ learned by `SymbolNet` and `PySR` analytically. `SymbolNet` slightly underestimates the extreme bins and shifts the missing events in these bins outside of $[-1, 1]$. This way the central regime of the distribution is accurately reproduced. `PySR`, with a formula restricted to $[-0.93, 0.93]$, dramatically fails in the extreme bins, leading to all central bins coming out high. Below, we show the correlation of learned and true CS-angles for all training events. For `SymbolNet`, the events closely follow the diagonal line, while the `PySR` formula leads to a small tilt. The bins with the highest number of events are at large $|\cos\theta^*|$, but for `PySR` they are shifted away from the diagonal, resulting in the observed underestimation of the extreme bins. The width of the colored area is smaller for `SymbolNet`, indicating a more accurate formula. `SymbolNet` also has more outliers away from the diagonal, which stems from the larger range of values the equation for scenario 6 allows. However, these are single events that can be attributed to

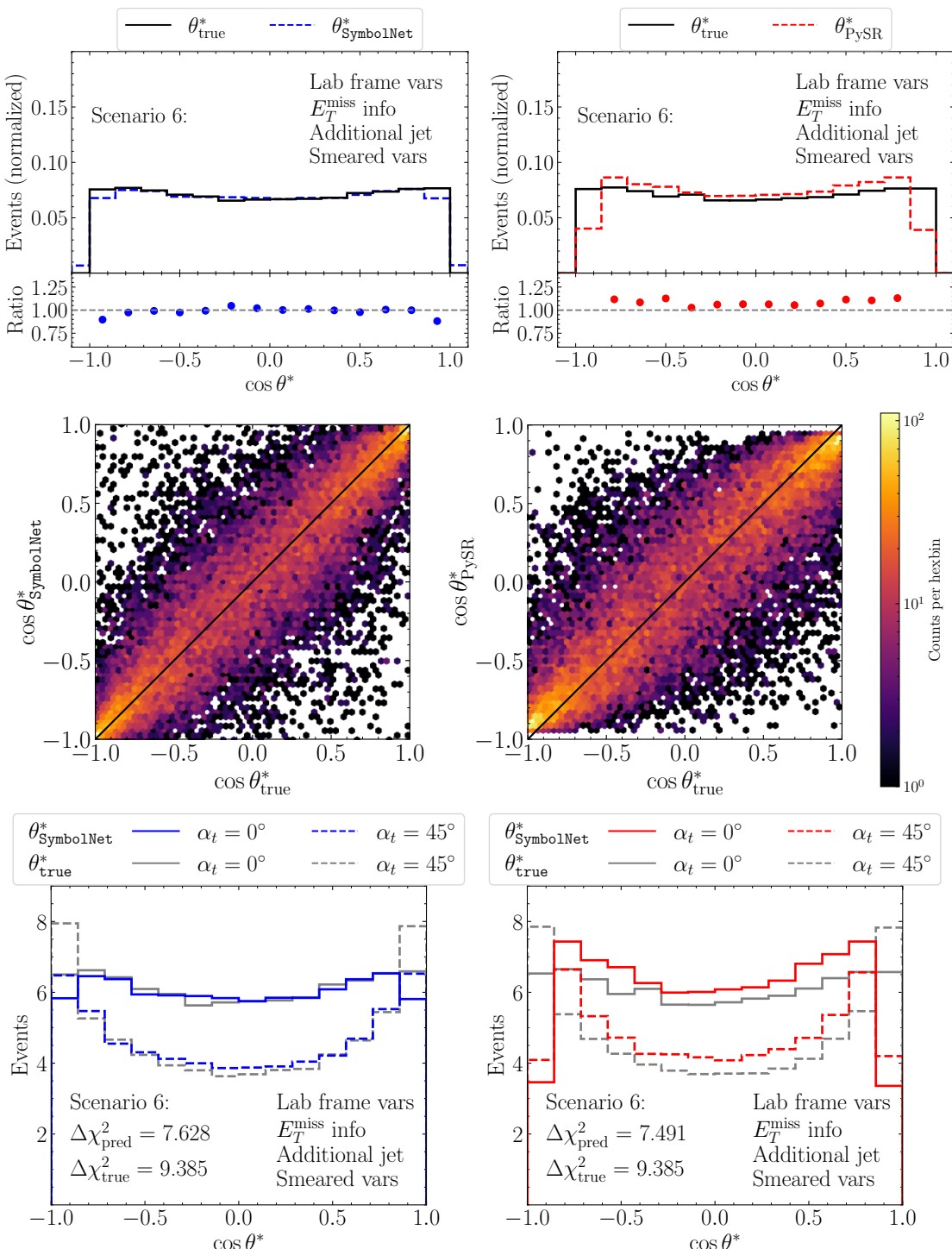

Figure 11: SymbolNet (left) and PySR (right) formulas for scenario 6 with perfect $b$-ordering. We show the learned CS angle, its calibration normalized to the same maximum count of events in one bin, and the distributions for $\alpha_t = 45°$ including the expected $\mathcal{CP}$-sensitivity.

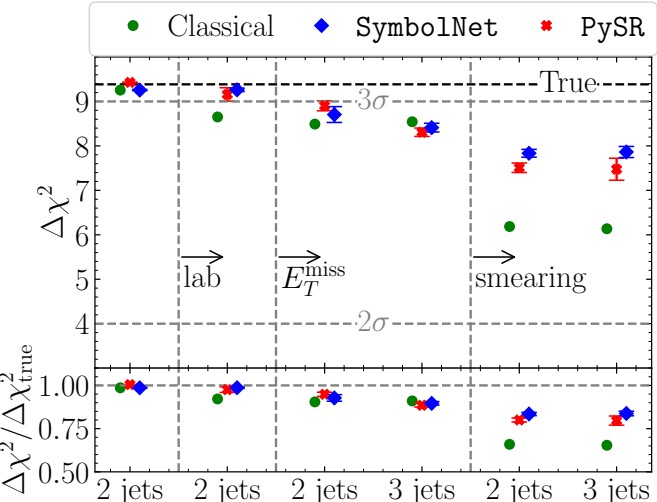

Figure 12: $\Delta\chi^2$ values for excluding $\alpha_t = 45°$ based on a measured SM-dataset. We show results for a classical reconstruction (green), PySR (red), and vectorized SymbolNet (blue), compared to the parton-level $\chi^2$. The central values and error bars correspond to the mean and the standard deviation of ten independent trainings.

statistical fluctuations.

In the lower panels in Fig. 11, we compare the $\mathcal{CP}$ sensitivity of the two learned formulas, testing $\alpha_t = 45°$ against the SM hypothesis $\alpha_t = 0°$. For this hypothesis test, the two formulas are trained on SM events. The number of expected events are taken from Ref. [60]. They are based on ATLAS analyses of the $t\bar{t}H$ channel and assume a luminosity of $\mathcal{L} = 300\text{fb}^{-1}$. For $\alpha_t = 45°$ the rate is smaller in the SM. Both learned formulas capture the main feature of the 45° case, namely that the CS distribution develops clear maxima for the extreme bins. Distinguishing the two datasets, SymbolNet reaches a sensitivity of $\Delta\chi^2_{\text{SymbolNet}} = 7.628$, compared to $\Delta\chi^2_{\text{PySR}} = 7.491$ and the parton level sensitivity $\Delta\chi^2_{\text{true}} = 9.385$.

Fig. 12 shows the $\Delta\chi^2$ values in each scenario for SymbolNet and PySR and compares them to the classical reconstruction algorithm. In scenario 1, all three methods get very close to the true parton-level information. The slightly lower results are an effect of MadSpin introducing small fluctuations in the decayed particle momenta. For the classical reconstruction, this effect becomes more drastic when the reconstructed top quarks are boosted. On the other hand, PySR and SymbolNet manage again to yield $\Delta\chi^2$ values close to the parton level case. In scenarios 3 and 4, the $\mathcal{CP}$ information drops slightly for all methods, but still yields similar results. The picture changes with detector smearing, where the classical reconstruction algorithm only captures around 60% of the total $\mathcal{CP}$ information. PySR and SymbolNet restore up to 80% of the $\mathcal{CP}$ information in these realistic scenarios, with SymbolNet giving slightly better results.

We furthermore investigate the performance of our methods for different $\mathcal{CP}$ angles in Fig. 13. We show $\Delta\chi^2$ from PySR, SymbolNet, and the classical reconstruction algorithms in scenario 6 for $\alpha_t = [10°, 25°, 45°, 90°]$. The results for $\alpha_t = 45°$ correspond to our above discussion. For the fully $\mathcal{CP}$-odd choice $\alpha_t = 90°$, the same relative $\mathcal{CP}$ information of $\theta^*$ is extracted by all three methods. For smaller angles, the performance of all methods drops, because of the less pronounced differences to the SM-kinematics and the greater impact of the smearing. Still, SymbolNet and PySR show a clear advantage over the classical reconstruction algorithm.

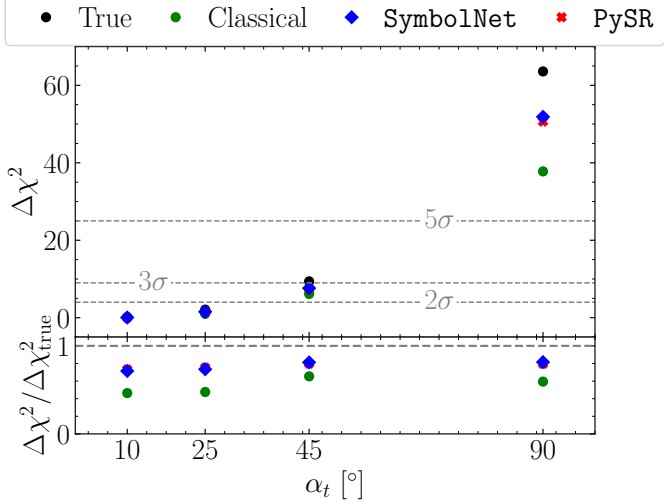

Figure 13: $\Delta\chi^2$ values from the PySR and SymbolNet formulas for scenario 6 by comparing the SM $\alpha_t = 0°$ hypothesis with different amounts of $\mathcal{CP}$ violation.

Finally, we again fix $\alpha_t = 45°$ and examine the data efficiency of PySR and SymbolNet in Fig. 14. Here, we compare the default PySR setup to two different setups of SymbolNet: The vectorized one, as used for the results in this chapter, and a scalar one, corresponding to the default setup in Ref. [32]. The gray dashed line indicates the information encoded in the total rate difference between the $\alpha_t = 0°$ and $\alpha_t = 45°$ hypotheses. PySR and the scalar SymbolNet reach a similar plateau for sufficient training data, with PySR performing better for small $N_{\text{train}}$. While this comparison shows a similar trend as Fig. 8, the additional 4-vector structure given to SymbolNet in the vectorized setup boosts its performance. This holds even for sparse training data, where it now outperforms PySR.

## 5   Conclusions

Searching for $\mathcal{CP}$ violation in the Higgs sector is a well-motivated but challenging task. Modern ML has been applied with great success to improve these searches. Typical numerical approaches, however, lack the interpretability and control needed to test a fundamental symmetry like $\mathcal{CP}$. We have shown how interpretable and controlled analytical expressions for testing the $\mathcal{CP}$ nature of the Higgs boson can be obtained using symbolic regression. We employed two complementary SR approaches: PySR based on a genetic algorithm, starting with a simple expression which is evolved into a more complicated expression; and SymbolNet based on a neural-network-like approach, starting with a complicated expression which is then successively reduced to a simpler expression.

In the first part of the paper, we have shown how to derive optimal $\mathcal{CP}$-odd observables at the detector level in analytic form. Focusing on VBF Higgs production, we compared the $\mathcal{CP}$ sensitivity of the observables learned using PySR and SymbolNet to a numerical BDT approach. We find the observables to outperform the BDT. We also showed that the SR approaches — in particular PySR— are more data efficient, meaning that they are able to analyze data with a very small $\mathcal{CP}$-odd component. Investigating the learned analytical expressions, we were able to confirm their $\mathcal{CP}$-odd nature explicitly. As expected, the learned $\mathcal{CP}$-odd observables have a similar structure as the known parton-level optimal observable. Our learned analytical optimal observables are straightforward to incorporate into actual analyses.

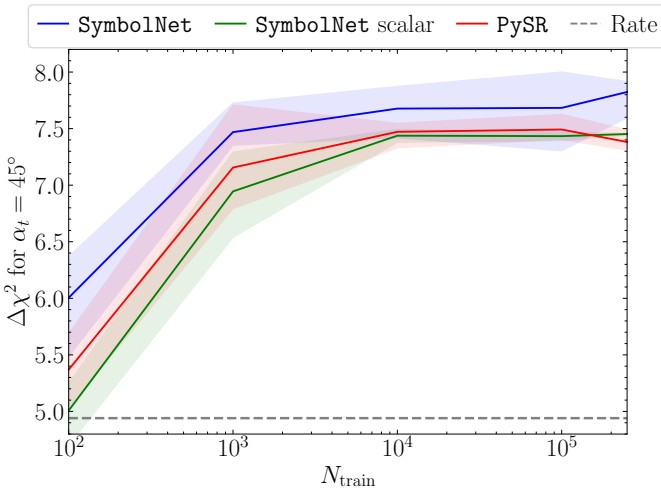

Figure 14: Test significance as a function of the training statistics for excluding the $\alpha_t = 45°$ hypothesis in scenario 6. The label "scalar" indicates the scalar setup of `SymbolNet` as opposed to the vectorized setup used throughout Sec. 4. The curves are averaged over four runs, with the bands indicating the minimum and maximum performance. The gray line indicates the minimal test significance due to total rate information.

In the second part of the paper, we focused on $t\bar{t}H$ production, where $\mathcal{CP}$-odd observables are very hard to measure at the LHC. We discussed the reconstruction of the $\mathcal{CP}$-sensitive parton-level Collins-Soper angle from reco-level data from semileptonic top decays. We studied six scenarios with different levels of available information, such as the neutrino momentum and limited detector resolution. In the most realistic scenarios, we found the SR approaches to be able to reconstruct the parton-level CS angle significantly better than traditional top-reconstruction algorithms. We also showed that the reconstructed CS angle preserves the $\mathcal{CP}$ sensitivity of the true parton-level CS angle. Due to the better expressivity, `SymbolNet` performs slightly better than PySR. Looking at the learned analytic expressions, we were able to identify structures similar to the parton-level CS angle. As for the optimal $\mathcal{CP}$-odd observables, the learned analytic approximations of the parton-level CS angle can be directly incorporated into the analysis of real data.

By comparing the PySR and scalar `SymbolNet` approaches, we found that PySR performs better if only a small amount of training data is available, while `SymbolNet` reaches similar or higher performance when a large amount of data is available. This demonstrates the complementarity of both methods. If physics information is provided to `SymbolNet` via a 4-vector structure, it can outperform PySR even for a small amount of data, demonstrating the power of such approaches. Our findings show that using SR for $\mathcal{CP}$ analyses not only strengthens the link between fundamental theory and complex experimental analyses, but it can also offer performance advantages in particular if data efficiency is important.

We expect that further applications of SR to $\mathcal{CP}$ violation in other processes can be realized straightforwardly and that it may also be used equivalently for the examination of other symmetries. Further enhancements to SR algorithms may be possible, e.g., by incorporating additional scientific knowledge beyond structure information into the algorithm [64].

# Acknowledgements

HB and TP acknowledge support through the KISS consortium (05D2022) funded by the German Federal Ministry of Education and Research BMBF in the ErUM-Data action plan, by the Deutsche Forschungsgemeinschaft (DFG, German Research Foundation) under grant 396021762 – TRR 257: Particle Physics Phenomenology after the Higgs Discovery, and through Germany's Excellence Strategy EXC 2181/1 – 390900948 (the Heidelberg STRUCTURES Excellence Cluster) and by the state of Baden-Württemberg through bwHPC and the German Research Foundation (DFG) through grant INST 35/1597-1 FUGG. EF and MM were funded by the Deutsche Forschungsgemeinschaft (DFG, German Research Foundation) under Germany's Excellence Strategy – EXC-2123 QuantumFrontiers – 390837967. The authors acknowledge resources provided by the LUIS computing cluster at Leibniz University Hannover, which is funded by the Deutsche Forschungsgemeinschaft (DFG, German Research Foundation) – Projektnummern INST 187/742-1 FUGG, INST 187/592-1 FUGG, and INST 187/430-1. This work has been partially funded by the Deutsche Forschungsgemeinschaft (DFG, German Research Foundation) - 491245950.

# A Building $\mathcal{CP}$-odd observables in $t\bar{t}H$

Dedicated $\mathcal{CP}$-odd measurements in $t\bar{t}H$ are not feasible for current studies at the LHC due to the smallness of the interference term. Despite, or precisely because of this, it is still interesting to test how much sensitivity an optimal $\mathcal{CP}$-odd observable can reach, as this is the only way of unambiguously testing for $\mathcal{CP}$ violation. The discussion in Section 3.1 still holds for $t\bar{t}H$, however, we use a slightly different SR algorithm.

$\mathcal{CP}$-odd observables can generally be defined via

$$\epsilon_{\mu\nu\rho\sigma}p_1^{\mu}p_2^{\nu}p_3^{\rho}p_4^{\sigma} \tag{46}$$

where the $p_i$ are linearly independent momenta or polarization vectors of the final or initial state particles. When boosted to the rest frame of one of the momenta, the $\epsilon$-tensors reduce to triple products (TP), which are commonly used in analyses targeting the interference term in $t\bar{t}H$ production (see e.g. [65–69]). For example, in the $t\bar{t}$ rest frame this results in [58,70]

$$\Delta\phi_{l\bar{l}}^{t\bar{t}} = \mathrm{sgn}[\vec{p}_t(\vec{p}_{l+} \times \vec{p}_{l-})] \ \arccos\left[\frac{\vec{p}_t \times \vec{p}_{l+}}{|\vec{p}_t \times \vec{p}_{l+}|} \cdot \frac{\vec{p}_t \times \vec{p}_{l-}}{|\vec{p}_t \times \vec{p}_{l-}|}\right]. \tag{47}$$

In particular, in the fully leptonic $t\bar{t}H$ decay, a set of 22 variables, dubbed $\omega_i$, can be defined out of combinations of the final state momenta [12,71]. From here on, we will refer to the $\epsilon$-tensors as TPs.

Here, we work in the fully leptonic decay channel but do not fix the momenta in Eq. (46). Instead, we let SymbolNet learn $\mathcal{CP}$-odd observables by construction using TPs:

- We define two instances of SymbolNet which do not share connections but inherit their input from the same input layer.
- The first instance of SymbolNet only contains $\mathcal{CP}$-even variables and is therefore $\mathcal{CP}$-even by construction. We call the output of this instance $D_{\mathrm{even}}(x)$.
- The second instance of SymbolNet has a special $V \to S$ layer, which only includes a TP. It is neither a unary, nor a binary operator and cannot be disabled during the training. All other functions in the remaining layers are $\mathcal{CP}$-even. Since all terms in the output must contain the TP exactly once, it is $\mathcal{CP}$-odd by construction. We refer to this part $D_{\mathrm{odd}}(x)$.
- Finally, we combine the two instances in a multiply layer and obtain $D(x) = D_{\mathrm{even}}(x) \cdot D_{\mathrm{odd}}(x)$ from which the optimal observable is constructed according to Eq. (23).

First, we show in Fig. 15 the distributions of the two TPs which yield the best $\mathcal{CP}$-sensitivities at parton level and detector level, respectively. The momenta in Eq. (46) are built from the Higgs momentum $p_h$, as well as even $(p_i + \bar{p}_i)$ and odd $(p_i - \bar{p}_i)$ combinations of the final state particles and their antiparticles. At parton level, the undecayed top quarks are included. This leads to

$$\epsilon_{\mathrm{parton}} = \epsilon_{\mu\nu\rho\sigma}(p_t + p_{\bar{t}})^{\mu}(p_t - p_{\bar{t}})^{\nu}(p_l + p_{\bar{l}})^{\rho}(p_l - p_{\bar{l}})^{\sigma} \tag{48}$$

being the best TP on parton level (from which the $\Delta\phi_{ll}^{t\bar{t}}$ can be derived). This is expected because the top quarks transfer their polarization information to their decay products and the leptons hold the maximal spin analyzing power [72,73]. At the detector level (see Sec. 4),

$$\epsilon_{\mathrm{reco}} = \epsilon_{\mu\nu\rho\sigma}(p_b + p_{\bar{b}})^{\mu}(p_b - p_{\bar{b}})^{\nu}(p_l + p_{\bar{l}})^{\rho}(p_l - p_{\bar{l}})^{\sigma} \tag{49}$$

yields the highest $\mathcal{CP}$ sensitivity. Here, we assume that the bottom and anti-bottom can be distinguished. This is experimentally very difficult. Our goal is, however, to give an optimistic estimate for the sensitivity of $\mathcal{CP}$-odd observables in $t\bar{t}H$ production.

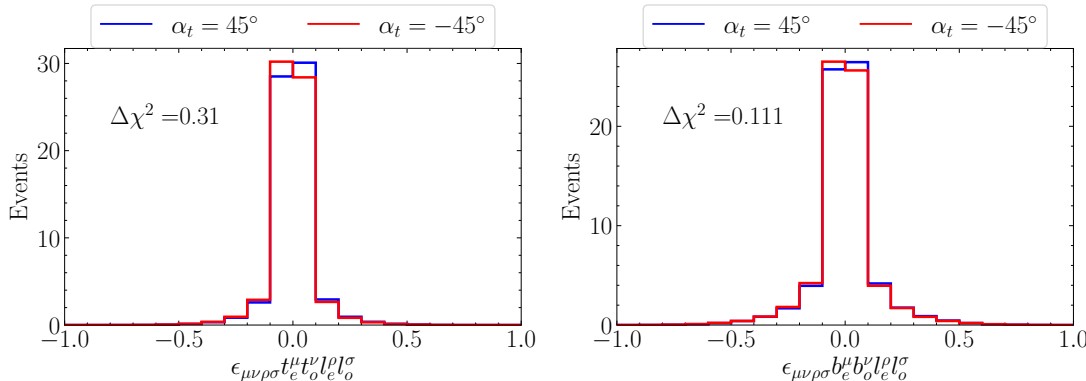

Figure 15: Distributions of $\epsilon_{\mu\nu\rho\sigma}(p_t+p_{\bar{t}})^{\mu}(p_t-p_{\bar{t}})^{\nu}(p_l+p_{\bar{l}})^{\rho}(p_l-p_{\bar{l}})^{\sigma}$ on parton level (left) and $\epsilon_{\mu\nu\rho\sigma}(p_b+p_{\bar{b}})^{\mu}(p_b-p_{\bar{b}})^{\nu}(p_l+p_{\bar{l}})^{\rho}(p_l-p_{\bar{l}})^{\sigma}$ on detector level (right) for $\alpha_t=\pm45°$.

Next, we train `SymbolNet` with the detector-level events to construct an optimal $\mathcal{CP}$-odd observable following the strategy outlined above. We use exactly one $V \to S$ and one $S \to S$ layer in both the $\mathcal{CP}$-even and $\mathcal{CP}$-odd part of the network, a learning rate of 0.005 and a batch size of 256. The result is shown in Fig. 16. The final equation after sparsity training consists of

$$
\begin{aligned}
D_{\text{odd}}(x) = \epsilon_{\mu\nu\rho\sigma} & \left(0.32 p_H + 0.622 p_b - 0.164 p_{\bar{l}} + 0.634 p_l\right)^{\mu} \\
& \left(0.298 p_H + 0.196 p_b + 0.54 p_{\bar{b}} + 0.551 p_{\bar{l}} - 0.216 p_l\right)^{\nu} \\
& \left(+0.259 p_H + 0.448 p_b - 0.319 p_{\bar{l}} + 0.761 p_l\right)^{\rho} \\
& \left(0.176 p_H - 0.051 p_b + 0.488 p_{\bar{b}} + 0.772 p_{\bar{l}} - 0.462 p_l\right)^{\sigma}
\end{aligned} \tag{50}
$$

and

$$
\begin{aligned}
D_{\text{even}}(x) = \text{Abs}\bigg( & \left(-0.123 p_{y,H} + 0.139 p_{y,b} + 0.159 p_{y,\bar{b}} - 0.375 p_{y,l} + 0.394 p_{z,H}\right. \\
& + 1.405 p_{z,b} - 0.134 p_{z,\bar{b}} - 0.627 p_{z,\bar{l}} + 0.803 p_{z,l} \\
& + 0.232 \left\| -2.715 p_b - 4.855 p_{\bar{b}} - 2.711 p_{\bar{l}} \right\|_3 \\
& + 0.016 \big\langle \left(4.888 p_H - 3.127 p_b + 4.287 p_{\bar{b}} - 1.211 p_{\bar{l}} - 2.628 p_l\right) \\
& \times \left(-2.516 p_H + 0.7615 p_b - 3.161 p_{\bar{b}} + 6.152 p_{\bar{l}} + 0.9251 p_l\right)\big\rangle_3 \bigg)^{1/2}.
\end{aligned} \tag{51}
$$

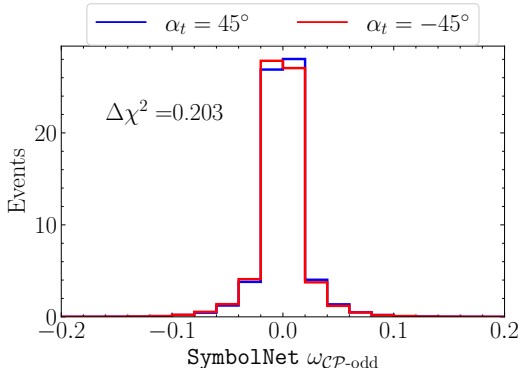

Figure 16: Distribution obtained from the equation predicted by `SymbolNet`.

As the $\Delta\chi^2$ values of the distributions show, there is a big jump in sensitivity from a single TP to the output of `SymbolNet`. Compared to the best parton-level TP, the best reco-level TP only reaches 36% of the total sensitivity, while `SymbolNet` reaches 65%. Over different training runs and hyperparameter settings, we observed that $D_{\mathrm{odd}}(x)$ often collapses to a single, linear TP. We find equations containing non-linear terms in the TP to not lead to higher $\Delta\chi^2$ values.

## B  Test statistics for asymmetry

The uncertainty per asymmetry bin can be obtained by propagating the Poisson errors for $N_i^+$ and $N_i^-$:

$$\sigma_{\mathcal{A}_i} = 2\sqrt{\frac{\left(N_i^-\sigma_{N_i^+}\right)^2 + \left(N_i^+\sigma_{N_i^-}\right)^2}{(N_i^+ + N_i^-)^4}} = 2\sqrt{\frac{N_i^+ \cdot N_i^-}{(N_i^+ + N_i^-)^3}} \, . \tag{52}$$

where the is obtained for $\sigma_{N_i^\pm} = \sqrt{N_i^\pm}$.

We then approximate the likelihood of $\mathcal{A}_i$ as a normal distribution $\mathcal{A}$ with mean $\mathcal{A}_i$ and width $\sigma_{\mathcal{A}_i}$. Thus, we can write the negative log-likelihood ratio between a BSM point and the SM as

$$q(\theta) = -2\log\frac{p(\{x\}|\theta)}{p(\{x\}|0)} = \sum_i \left[ \frac{(\mathcal{A}_i(\theta) - \mathcal{A}_i(0))^2}{\sigma_{\mathcal{A}_i}^2(\theta)} + \log\frac{\sigma_{\mathcal{A}_i}^2(\theta)}{\sigma_{\mathcal{A}_i}^2(0)} \right] , \tag{53}$$

where we assume that the observed dataset $\{x\}$ follows the SM expectations — i.e. drawn from the likelihood $p(x|\theta)$.

Following Wilks' theorem, we assume the test statistic $q$ to follow a $\chi^2$ distribution. In our analysis, we exclude all bins with less than three events.

## C  Loss function for Collin-Soper angle reconstruction

One of the main challenges in most LHC analyses is to reconstruct the parton-level information from events obtained at the reconstruction level such that

$$p_{\mathrm{reco}}(x|\theta) = \int dx_{\mathrm{parton}} \, p(x_{\mathrm{reco}}|x_{\mathrm{parton}}) \, p(x_{\mathrm{parton}}|\theta) \, . \tag{54}$$

This reco-level likelihood for a vector of parameters of interest $\theta$ depends on the parton level events via $p(x_{\mathrm{reco}}|x_{\mathrm{parton}})$. This conditional probability includes parton showering, hadronization, detector resolution, and any other effects that might appear.

Our SR algorithms need to be able to approximate $p(x_{\mathrm{reco}}|x_{\mathrm{parton}})$. This raises the question of the optimal objective function for such a task. In a regression task with Gaussian errors on the fit parameters, we can show

$$-\log\mathcal{L}_{\mathrm{Normal}} \sim \sum_i (y_i - \hat{y}_i)^2 \tag{55}$$

for predicted (true) values $\hat{y}_i$ ($y_i$). Therefore, the MSE loss is optimal for solving such a problem. However, Eq. (54) includes much more than just a Gaussian smearing. For example,

neutrino information is lost and jets may be wrongly reconstructed or identified. This means in practice that the training data contains outliers which can spoil the convergence of the SR algorithm. To avoid this, we can adapt the MSE loss function.

We obtained the best results using an "inverse Gaussian loss"

$$\mathcal{L}_{\text{InvGaussian}} = 1 - \exp\left(-\frac{(y - \hat{y})^2}{2\left(\frac{\max(y)}{\sigma} - \frac{\min(y)}{\sigma}\right)^2}\right). \tag{56}$$

For $y \sim \hat{y}$, $\mathcal{L}_{\text{InvGaussian}}$ collapses to the MSE loss, while for $|y - \hat{y}| \gg 0$ it approaches one, making it robust against outliers. $\sigma$ can be used to tune the loss to the expected range of predictions.

# D  Formulas for the Collins-Soper angle

Here, we present the formulas that were found by `PySR` and `SymbolNet` for the reconstruction of the CS angle $\cos\theta^*$. Individual components of the momenta are labeled by $E$, $p_x$, $p_y$, and $p_z$. Norms and dot products of 4-vectors correspond to the Minkowski norm and product, respectively.

**Scenario 1:**

`PySR`:

$$\cos\theta^*_{\text{PySR}} = \frac{p_{z,b} + p_{z,\bar{l}} + p_{z,\nu}}{\sqrt{\left(p_{x,b} + p_{x,\bar{l}} + p_{x,\nu}\right)^2 + \left(p_{y,b} + p_{y,\bar{l}} + p_{y,\nu}\right)^2 + \left(p_{z,b} + p_{z,\bar{l}} + p_{z,\nu}\right)^2}}$$

`SymbolNet`:

$$\cos\theta^*_{\text{SymbolNet}} = \frac{1.006 p_{z,b} + 1.001 p_{z,\bar{l}} + 1.002 p_{z,\nu} - 1.027 p_{z,\bar{b}} - 1.027 p_{z,q} - 1.031 p_{z,\bar{q}}}{\left\| 1.034 p_b + 1.022 p_{\bar{l}} + 1.024 p_\nu - p_{\bar{b}} - 1.007 p_q - 1.009 p_{\bar{q}} \right\|_3}$$

**Scenario 2:**

`PySR`:

$$\cos\theta^*_{\text{PySR}} = \sin\left[1.41\left(p_{z,b} + 1.41 p_{z,\bar{l}} + 1.41 p_{z,\nu} - 1.41 p_{z,\bar{b}} - 1.41 p_{z,q} - 1.41 p_{z,\bar{q}}\right)\middle/\right.$$
$$\left(E_b + E_{\bar{l}} + E_\nu + E_{\bar{b}} + E_q + E_{\bar{q}}\right.$$
$$\left.\left. + \sqrt{\left(-p_{y,b} - p_{y,\nu} + p_{y,\bar{b}} + p_{y,\bar{q}}\right)^2 + \left(p_{x,b} + p_{x,\bar{l}} + p_{x,\nu} - p_{x,\bar{b}} - p_{x,\bar{q}}\right)^2} - 1.70\right)\right]$$

`SymbolNet`:

$$
\begin{aligned}
\cos\theta^*_{\texttt{SymbolNet}} = & 1.006\Bigg(\text{boost}\Big[-2.793p_b - 2.811p_{\bar{l}} - 2.793p_\nu + 2.806p_{\bar{b}} + 2.835p_q + 2.815p_{\bar{q}}\Big| \\
& -2.443p_b - 2.434p_{\bar{l}} - 2.456p_\nu - 2.45p_{\bar{b}} - 2.429p_q - 2.398p_{\bar{q}}\Big]\Big)\Big|_z \Bigg/ \\
& \Bigg\|\text{boost}\Big[-2.793p_b - 2.811p_{\bar{l}} - 2.793p_\nu + 2.806p_{\bar{b}} + 2.835p_q + 2.815p_{\bar{q}}\Big| \\
& -2.443p_b - 2.434p_{\bar{l}} - 2.456p_\nu - 2.45p_{\bar{b}} - 2.429p_q - 2.398p_{\bar{q}}\Big]\Bigg\|_3
\end{aligned}
$$

**Scenario 3:**

PySR:

$$
\begin{aligned}
\cos\theta^*_{\text{PySR}} = \sin\Bigg[ & \Big(-0.300\sqrt{E_T^{\text{miss}}}\,(p_{z,b} + 2.22p_{z,\bar{l}})\big(p_{y,\bar{l}} + p_{x,b} + p_{x,\bar{l}} - 1.99\big) \\
& + p_{z,b} + p_{z,\bar{l}} - p_{z,\bar{b}} - p_{z,q} - p_{z,\bar{q}}\Big)\Bigg/ \\
& \Big(E_T^{\text{miss}} + 0.949E_b + 0.949E_{\bar{b}} + E_{\bar{l}} + E_q + 0.949E_{\bar{q}} - 1.12\Big)\Bigg]
\end{aligned}
$$

`SymbolNet`:

$$
\begin{aligned}
\cos\theta^*_{\texttt{SymbolNet}} = & -1.079\Bigg(\text{boost}\Big[-4.831p_b - 4.919p_{\bar{l}} - 0.2614p^{\text{miss}} + 4.752p_{\bar{b}} + 4.827p_q + 4.711p_{\bar{q}}\Big| \\
& 4.506p_b + 20.27p_{\bar{l}} + 8.622p_{\bar{b}} + 10.6p_q + 8.935p_{\bar{q}}\Big] \\
& -0.698\tanh\big(0.711p_b + 0.821p_{\bar{l}} + 2.083p^{\text{miss}} - 0.717p_{\bar{b}} - 0.575p_q - 0.672p_{\bar{q}}\big)\Big)\Big|_z\Bigg/ \\
& \Bigg(\Bigg\|\text{boost}\Big[-4.831p_b - 4.919p_{\bar{l}} - 0.2614p^{\text{miss}} + 4.752p_{\bar{b}} + 4.827p_q + 4.711p_{\bar{q}}\Big| \\
& 4.506p_b + 20.27p_{\bar{l}} + 8.622p_{\bar{b}} + 10.6p_q + 8.935p_{\bar{q}}\Big]\Bigg\|_3 \\
& + 1.039\Big\|\tanh\big(0.711p_b + 0.821p_{\bar{l}} + 2.083p^{\text{miss}} - 0.717p_{\bar{b}} - 0.575p_q - 0.672p_{\bar{q}}\big)\Big\|_3^2 + 0.089\Bigg)
\end{aligned}
$$

**Scenario 4:**

PySR:

$$
\begin{aligned}
\cos\theta^*_{\text{PySR}} = \sin\Bigg[ & \Big(-p_{z,\bar{b}} - p_{z,q} - p_{z,\bar{q}} + 1.06\sqrt{0.888E_T^{\text{miss}} + 1}\,(p_{z,b} + 1.24p_{z,\bar{l}})\Big)\Bigg/ \\
& \Big(E_T^{\text{miss}} + 0.968E_b + 0.826E_{\bar{b}} + E_{\bar{l}} + p_{y,\bar{l}}^2 + E_q + E_{\bar{q}} \\
& + 0.494\big(-p_{x,b} - p_{x,\bar{l}} + p_{x,\bar{b}} + p_{x,\bar{q}}\big) - 1.48\Big)\Bigg]
\end{aligned}
$$

SymbolNet:

$$\cos\theta^*_{\texttt{SymbolNet}} = 1.074\Bigg(\text{boost}\Big[-4.341p_b - 4.71p_{\bar{l}} - 0.2542p^{\text{miss}} + 4.087p_{\bar{b}} + 4.28p_q + 4.133p_{\bar{q}}\Big|$$

$$-4.103p_b - 14.52p_{\bar{l}} - 7.11p_{\bar{b}} - 7.261p_q - 6.894p_{\bar{q}} - 0.327\Big]$$

$$+ 0.668\tanh\big(0.672p_b + 0.723p_{\bar{l}} + 1.949p^{\text{miss}} - 0.624p_{\bar{b}} - 0.625p_q - 0.62p_{\bar{q}}\big)\Big)\Big|_z\Bigg/$$

$$\Bigg(\Big\|\text{boost}\Big[-4.341p_b - 4.71p_{\bar{l}} - 0.2542p^{\text{miss}} + 4.087p_{\bar{b}} + 4.28p_q + 4.133p_{\bar{q}}\Big|$$

$$-4.103p_b - 14.52p_{\bar{l}} - 7.11p_{\bar{b}} - 7.261p_q - 6.894p_{\bar{q}} - 0.327\Big]$$

$$+ 0.162\tanh\big(0.672p_b + 0.723p_{\bar{l}} + 1.949p^{\text{miss}} - 0.624p_{\bar{b}} - 0.625p_q - 0.62p_{\bar{q}}\big)\Big\|_3$$

$$0.906\Big\|\tanh\big(0.672p_b + 0.723p_{\bar{l}} + 1.949p^{\text{miss}} - 0.624p_{\bar{b}} - 0.625p_q - 0.62p_{\bar{q}}\big)\Big\|_3^2 + 0.154\Bigg)$$

**Scenario 5:**

PySR:

$$\cos\theta^*_{\texttt{PySR}} = \sin\Bigg[\Bigg(1.156p_{z,b} - \frac{1.156p_{z,\bar{b}}}{E_T^{\text{miss}}\big(0.710p_{x,\bar{b}} + p_{x,q} + p_{x,\bar{q}}\big) + 0.892}$$

$$+ 2.06p_{z,\bar{l}} - 1.156p_{z,q} - 1.312p_{z,\bar{q}}\Bigg)\Bigg/$$

$$\Bigg(E_b + E_{\bar{b}} - \sqrt{E_{\bar{l}}} + 2E_{\bar{l}} + p_{x,\bar{l}}^2 + E_q + E_{\bar{q}} + p_{y,\bar{q}}^2 + \big(-p_{y,\bar{l}} + p_{y,q}\big) - 0.511\Bigg)\Bigg]$$

SymbolNet:

$$\cos\theta^*_{\texttt{SymbolNet}} = -0.928\Bigg(\text{boost}\Big[1.673p_b + 2.866p_{\bar{l}} - 1.666p_{\bar{b}} - 2.495p_q - 2.294p_{\bar{q}}\Big|$$

$$-3.822p_b - 5.112p_{\bar{l}} - 3.246p_{\bar{b}} - 3.755p_q - 3.459p_{\bar{q}}\Big]$$

$$-0.131\tanh\big(0.8275p_{\bar{l}} - 0.3036p_{\bar{q}}\big)\Big)\Big|_z\Bigg/$$

$$\Big\|\text{boost}\Big[1.673p_b + 2.866p_{\bar{l}} - 1.666p_{\bar{b}} - 2.495p_q - 2.294p_{\bar{q}}\Big|$$

$$-3.822p_b - 5.112p_{\bar{l}} - 3.246p_{\bar{b}} - 3.755p_q - 3.459p_{\bar{q}}\Big]\Big\|_3$$

**Scenario 6:**

PySR:

$$\cos\theta^*_{\text{PySR}} = \sin\left[\left(1.114 p_{z,b} + 2.143 p_{z,\bar{l}} - 0.858 p_{z,\bar{b}} - 0.426 p_{z,q} - 1.088 p_{z,\bar{q}}\right)\bigg/\right.$$
$$\left(E_T^{\text{miss}} E_q + E_b + E_{\bar{b}} + 1.85 E_{\bar{l}} + E_{\bar{q}}\right.$$
$$\left.\left.-\left(p_{x,b} + p_{x,\bar{l}}\right)\left(p_{x,\bar{b}} + p_{x,q} + p_{x,\bar{q}}\right) - 0.863 - \frac{0.205 p_{z,q}}{\sqrt{E_q}}\right)\right]$$

SymbolNet:

$$\cos\theta^*_{\text{SymbolNet}} = 0.971\left(\text{boost}\left[-1.521 p_b - 3.066 p_{\bar{l}} + 1.421 p_{\bar{b}} + 2.218 p_q + 2.043 p_{\bar{q}}\right|\right.$$
$$-2.845 p_b - 4.559 p_{\bar{l}} - 2.575 p_{\bar{b}} - 2.753 p_q - 2.676 p_{\bar{q}}\right]$$
$$\left.-0.256 \tanh\left(0.646 p_{\bar{b}} - 0.634 p_{\bar{l}} + 0.648 p_q\right)\right)\bigg|_z \bigg/$$
$$\left(\left\|\text{boost}\left[-1.521 p_b - 3.066 p_{\bar{l}} + 1.421 p_{\bar{b}} + 2.218 p_q + 2.043 p_{\bar{q}}\right|\right.\right.$$
$$\left.-2.845 p_b - 4.559 p_{\bar{l}} - 2.575 p_{\bar{b}} - 2.753 p_q - 2.676 p_{\bar{q}}\right]\bigg\|_3$$
$$+0.774\left\langle 0.071 \,\text{boost}\left[-1.521 p_b - 3.066 p_{\bar{l}} + 1.421 p_{\bar{b}} + 2.218 p_q + 2.043 p_{\bar{q}}\right|\right.$$
$$\left.-2.845 p_b - 4.559 p_{\bar{l}} - 2.575 p_{\bar{b}} - 2.753 p_q - 2.676 p_{\bar{q}}\right]$$
$$+\tanh\left(0.646 p_{\bar{b}} - 0.634 p_{\bar{l}} + 0.648 p_q\right)\times$$
$$\left.\left.\tanh\left(0.646 p_{\bar{b}} - 0.634 p_{\bar{l}} + 0.648 p_q\right)\right\rangle_3 - 0.223\right)$$

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
