# Peer review of "$\mathcal{CP}$-Analyses with Symbolic Regression"

_SciPost Physics_

## Round 1 · Referee Report · Joshua Lorne Bendavid (Referee 1) · 2025-10-9

Strengths

Application of relatively unused technique (symbolic regression) to relevant problems in particle physics.

Clear description of methodology

Weaknesses

Some missing details in the description (see detailed comments)

Conclusions should give some perspective on possible future work in this area

Report

The paper gives a clear description of the application of a relatively new technique in particle physics (symbolic regression) to relevant problems in the field, and in particular to problems which are well suited to the particular strengths of symbolic regression (test of CP structure in Higgs and Top production where the CP even/odd structure may be manifest in analytic formulae)

The results are promising for the performance of the symbolic regression methods, especially in cases where the available training sample size is limited.

I recommend that the paper is accepted for publication after the (modest) requested changes.

Requested changes

  1. title: Could be more informative/detailed, consider adding something about "Higgs", "top" or "collider phenomenology" or similar

Introduction

  1. consider add citation to https://arxiv.org/abs/2508.00989

Section 2.1

  1. "but poorly initialized." needs further explanation since the term initialization has not been introduced in the paper up to this point

Section 2.2

  1. Eq (7): Explain in the text the meaning of Theta (I assume that it's the heaviside function?)

  2. If the threshold parameter exceeds one and the heaviside function is removed, doesn't this produce null gradients with respect to the threshold parameter during the pruning step? Consider adding a statement about this in the text. (if the Theta function is replaced as in eq(12) and kept in the backpropagation for this case, then explain this)

  3. Fix inconsistent notation (capital vs lowercase) for threshold parameters in equations vs Table 1

  4. Suggest to add comment on the choice and sensitivity to the value kappa=5

Section 3.1

  1. "the CP-properties of d and omega_CP-odd are the same.": shouldn't this read "of D and omega_CP-odd are the same" ? (ie the comment is about the transformed quantity)

Section 3.2

  1. "at two tagging jets" -> "at least two tagging jets"? (or exactly two tagging jets?)

Section 3.3

  1. Better explanation needed on the meaning of Eq. 30 (the "#[] notation in particular")

  2. Eq (33) some explanation about what is different here with respect to (32) would be useful. Different random initialization of the weights?

  3. To facilitate comarison between Fig. 7 and Table. 2 suggest explicitly writing in the text that significance = sqrt(chi^2) in this context

Section 4.2

  1. Fig. 10 and surrounding discussion. Most likely the classical reconstruction used doesn't take explicitly into account the expected resolution of the different objects (e.g. from a kinematic fit) which is different for jets, leptons, and missing energy. It's encouraging that the symbolic regression is able to mitigate this (likely by effectively downweighting the less well-measured objects in the combinatin of kinematics). Some more explicit comment on this in the text may be worthwhile.

  2. "but these single events can be attributed to statistical fluctuations": this is probably also partly related to the restricted/narrower output range of the PySR-learned function which would tend to "compress" the output space compared to the function learned by SymbolNet

  3. Fig. 11 and surrounding discussion: it would be useful to show the comparison for alpha_t=0 vs alpha_t=45 degrees also for phi*_true

Conclusions

  1. Consider adding some comment on possible directions for follow-up work

Recommendation

Ask for minor revision

---

## Round 1 · Referee Report · Anonymous (Referee 2) · 2025-10-30

Strengths

1 well written paper with thorough analysis of the results 2 the explainability argument for symbolic regressing which is always put forward is convincingly supported by the analysis of the obtained models : this is very rarely the case, 3 : the performance is clearly better, in particular when the number of training events is low, which is a strong argument for applying these method in real experiments at the LHC

Weaknesses

1 comparison to a dense NN could be provided
2 learning curve for the second application is missing

Report

This papers applies symbolic regression to two CP sensitive observables. Two distinct symbolic algorithms are applied PySR and SymbolNet, the second one with improvements by the authors. For me the paper stands out for two reasons : first, the explainability argument for symbolic regressing which is always put forward is convincingly supported by the analysis of the obtained models : this is very rarely the case, second : the performance are clearly better, in particular when the number of training events is low, which is a strong argument for applying these method in real experiments at the LHC. The paper is also very well structured and written.

I support strongly this paper for publication, with minor modifications.

Requested changes

1 please run the paper through an advanced spell checker, I've spotted "mathmatical" there must be more

2 table 2 : you can bold face the largest value in each case, this is standard in ML paper to help the reader

3 it is great that you compare systematically with BDT. It would have been nice and convincing to compare in addition to a regular dense NN (which I expect to be at the level of the BDT). Actually if you stop the training of Symbolnet after the "default training" (section 2.2), is it not a regular dense NN ? Then you could use this as a NN reference and show it in the results and plots to see "in action" the symbolic layers.

4 I really really like Figure 8, can't you do a similar one for the Collin Soper angle case ?

5 A large amount of work went into SymbolNet, to have it "understand" 4-momenta. Given that I'm sure you were disappointed by the poorer performance (compared to PySR) for the WBF Higgs production case, and the marginally better one for Collin Soper angle case. Wouldn't it be possible to plug 4-momentum symbolic capabilities to PySR ?

6 Bibliography :
[15] : please add CERN report number CERN-LHCEFTWG-2022-001, CERN-LPCC-2022-05
[27] : very important, this is NOT a NeurIPS paper, this is a paper accepted in ML4PS workshop at NeurIPS which is much easier than proper NeurIPS. Please fix
[29] and beyond : 29 has been published in CSBS, please fix, and please check the following references that are only on arXiv
[30] something went wrong in formatting the title
[33] np-hard => NP-hard
[46] XGboost : I believe the standard way (if any) to cite xgboost is to cite the original publication as below, with arXiv:1603.02754 in addition (not twice as now in any case)

@inproceedings{Chen:2016:XST:2939672.2939785,
author = {Chen, Tianqi and Guestrin, Carlos},
title = {{XGBoost}: A Scalable Tree Boosting System},
booktitle = {Proceedings of the 22nd ACM SIGKDD International Conference on Knowledge Discovery and Data Mining},
series = {KDD '16},
year = {2016},
isbn = {978-1-4503-4232-2},
location = {San Francisco, California, USA},
pages = {785--794},
numpages = {10},
url = {http://doi.acm.org/10.1145/2939672.2939785},
doi = {10.1145/2939672.2939785},
acmid = {2939785},
publisher = {ACM},
address = {New York, NY, USA},
keywords = {large-scale machine learning},
}
[49] : this CMS citation is not consistent with the next ones

Recommendation

Publish (easily meets expectations and criteria for this Journal; among top 50%)

---

## Round 2 · Author Response

Dear Editor and Referees,

We thank the referees for their careful reviews of our manuscript and the helpful comments. We have addressed the various points raised in the referee's reports and modified the text accordingly to improve the clarity of our manuscript. For convenience, all changes in the text are marked in blue. Below, we answer in detail to the different points raised by the referees.

We hope that with these additions, our article can now be accepted for publication.

Yours sincerely,
H. Bahl, E. Fuchs, M. Menen, T. Plehn

---

## Round 2 · List of Changes

Referee 1

1 please run the paper through an advanced spell checker, I've spotted "mathmatical" there must be more

  • Done.

2 table 2 : you can bold face the largest value in each case, this is standard in ML paper to help the reader

  • We followed the suggestion.

3 it is great that you compare systematically with BDT. It would have been nice and convincing to compare in addition to a regular dense NN (which I expect to be at the level of the BDT). Actually if you stop the training of Symbolnet after the "default training" (section 2.2), is it not a regular dense NN ? Then you could use this as a NN reference and show it in the results and plots to see "in action" the symbolic layers.

  • As suggested, we trained SymbolNet observables without any sparsity loss. We have added the results to Tab. 2 and adapted the description in the text accordingly.

4 I really really like Figure 8, can't you do a similar one for the Collin Soper angle case ?

  • We have added the requested Figure (now Fig. 14 in the paper) and a corresponding description. We also slightly changed our conclusions based on the results in this Figure.

5 A large amount of work went into SymbolNet, to have it "understand" 4-momenta. Given that I'm sure you were disappointed by the poorer performance (compared to PySR) for the WBF Higgs production case, and the marginally better one for Collin Soper angle case. Wouldn't it be possible to plug 4-momentum symbolic capabilities to PySR ?

  • We considered this possibility. After looking into the PySR code, we, however, realized that adding vector support would require a larger reorganization of the code. In particular, PySR is build upon the Julia package SymbolicRegression which itself is based on the package DynamicExpressions. Adding vector support would require modifying each of these three packages. Since the required amount of work was out of scope for the project, we dismissed this idea. Implementing vector support into SymbolNet on the other hand turned out to be relatively straightforward since SymbolNet is a self-contained package with only a minimal set of application layers.

6 Bibliography : [15] : please add CERN report number CERN-LHCEFTWG-2022-001, CERN-LPCC-2022-05 [27] : very important, this is NOT a NeurIPS paper, this is a paper accepted in ML4PS workshop at NeurIPS which is much easier than proper NeurIPS. Please fix [29] and beyond : 29 has been published in CSBS, please fix, and please check the following references that are only on arXiv [30] something went wrong in formatting the title [33] np-hard => NP-hard [46] XGboost : I believe the standard way (if any) to cite xgboost is to cite the original publication as below, with arXiv:1603.02754 in addition (not twice as now in any case)

@inproceedings{Chen:2016:XST:2939672.2939785, author = {Chen, Tianqi and Guestrin, Carlos}, title = {{XGBoost}: A Scalable Tree Boosting System}, booktitle = {Proceedings of the 22nd ACM SIGKDD International Conference on Knowledge Discovery and Data Mining}, series = {KDD '16}, year = {2016}, isbn = {978-1-4503-4232-2}, location = {San Francisco, California, USA}, pages = {785--794}, numpages = {10}, url = {http://doi.acm.org/10.1145/2939672.2939785}, doi = {10.1145/2939672.2939785}, acmid = {2939785}, publisher = {ACM}, address = {New York, NY, USA}, keywords = {large-scale machine learning}, } [49] : this CMS citation is not consistent with the next ones

  • We fixed these issues.

Referee 2

title: Could be more informative/detailed, consider adding something about "Higgs", "top" or "collider phenomenology" or similar

  • While we agree that the focus of the paper is CP violation in the Higgs sector, the used methods can be straightforwardly applied to the search for CP violation in other sectors (as mentioned in the conclusions). We, therefore, would prefer keeping the more general title.

Introduction

consider add citation to https://arxiv.org/abs/2508.00989

  • We followed the suggestion.

Section 2.1

"but poorly initialized." needs further explanation since the term initialization has not been introduced in the paper up to this point

  • We have modified the sentence for improved clarity.

Section 2.2

Eq (7): Explain in the text the meaning of Theta (I assume that it's the heaviside function?)

  • We have added a comment to Eq (5) where it shows up for the first time.

If the threshold parameter exceeds one and the heaviside function is removed, doesn't this produce null gradients with respect to the threshold parameter during the pruning step? Consider adding a statement about this in the text. (if the Theta function is replaced as in eq(12) and kept in the backpropagation for this case, then explain this)

  • For the gradient computation during backpropagation, the theta function is indeed replaced by the function in Eq(12), which is non-zero everywhere and should not produce any null gradients. We have added a comment about the gradient stability.

Fix inconsistent notation (capital vs lowercase) for threshold parameters in equations vs Table 1

  • We have fixed this.

Suggest to add comment on the choice and sensitivity to the value kappa=5

  • We have cited the original SymbolNet paper as a reference for this value of kappa.

Section 3.1

"the CP-properties of d and omega_CP-odd are the same.": shouldn't this read "of D and omega_CP-odd are the same" ? (ie the comment is about the transformed quantity)

  • We have fixed this.

Section 3.2

"at two tagging jets" -> "at least two tagging jets"? (or exactly two tagging jets?)

  • At least two jets are required. We clarified this in the text.

Section 3.3

Better explanation needed on the meaning of Eq. 30 (the "#[] notation in particular")

  • We have added a clarifying statement.

Eq (33) some explanation about what is different here with respect to (32) would be useful. Different random initialization of the weights?

  • Yes, a different initialization is used. We clarified this in the text.

To facilitate comarison between Fig. 7 and Table. 2 suggest explicitly writing in the text that significance = sqrt(chi^2) in this context

  • We have added a sentence in the paragraph related to Tab. 2.

Section 4.2

Fig. 10 and surrounding discussion. Most likely the classical reconstruction used doesn't take explicitly into account the expected resolution of the different objects (e.g. from a kinematic fit) which is different for jets, leptons, and missing energy. It's encouraging that the symbolic regression is able to mitigate this (likely by effectively downweighting the less well-measured objects in the combinatin of kinematics). Some more explicit comment on this in the text may be worthwhile.

  • We agree that these results indicate a better handling of the different resolutions by the SR algorithms and have added a comment in the description of Fig. 10.

"but these single events can be attributed to statistical fluctuations": this is probably also partly related to the restricted/narrower output range of the PySR-learned function which would tend to "compress" the output space compared to the function learned by SymbolNet

  • We agree and have extended our statement about the outliers.

Fig. 11 and surrounding discussion: it would be useful to show the comparison for alpha_t=0 vs alpha_t=45 degrees also for phi*_true

  • The true distributions are now included in the respective plots.

Conclusions

Consider adding some comment on possible directions for follow-up work

  • We have added a small outlook paragraph.

---

## Editorial Decision

refereeing_in_preparation